# Cross-talk between individual phenol-soluble modulins in *Staphylococcus aureus* biofilm enables rapid and efficient amyloid formation

**Masihuz Zaman, Maria Andreasen\***

Aarhus University, Department of Biomedicine, Aarhus, Denmark

**Abstract** The infective ability of the opportunistic pathogen *Staphylococcus aureus*, recognized as the most frequent cause of biofilm-associated infections, is associated with biofilm-mediated resistance to host immune response. Phenol-soluble modulins (PSM) comprise the structural scaffold of *S. aureus* biofilms through self-assembly into functional amyloids, but the role of individual PSMs during biofilm formation remains poorly understood and the molecular pathways of PSM self-assembly are yet to be identified. Here we demonstrate high degree of cooperation between individual PSMs during functional amyloid formation. PSMα3 initiates the aggregation, forming unstable aggregates capable of seeding other PSMs resulting in stable amyloid structures. Using chemical kinetics we dissect the molecular mechanism of aggregation of individual PSMs showing that PSMα1, PSMα3 and PSMβ1 display secondary nucleation whereas PSMβ2 aggregates through primary nucleation and elongation. Our findings suggest that various PSMs have evolved to ensure fast and efficient biofilm formation through cooperation between individual peptides.

**\*For correspondence:**
mariaj@biomed.au.dk

**Competing interests:** The authors declare that no competing interests exist.

## Introduction

Aggregated proteins in the form of functional amyloids are widespread in nature (*Pham et al., 2014*). In humans, functional amyloids assist in immunity, reproduction, and hormone secretion (*Maji et al., 2009*). However, in various bacterial strains they provide structural stability as the major protein component of the self-produced polymeric matrix in biofilms (*Dueholm et al., 2010*; *Evans and Chapman, 2014*; *Romero et al., 2010*; *Schwartz et al., 2012*). Functional amyloids increase the bacteria's ability toward a variety of environmental insults, increasing their persistence in the host as well as promoting resistance to antimicrobial drugs and the immune system (*Gallo et al., 2015*; *Marmont et al., 2017*; *Van Gerven et al., 2018*). The well-studied curli machinery in *Escherichia coli* (*Evans and Chapman, 2014*), Fap system in *Pseudomonas fluorescens* (*Dueholm et al., 2010*), TasA system in *Bacillus subtilis* (*Romero et al., 2010*), along with phenol-soluble modulins (PSMs) in *Staphylococcus aureus* (*Schwartz et al., 2012*) are some of the major bacterial functional amyloid systems that have been reported so far.

For *S. aureus* biofilm formation PSMs have been recognized as a crucial factor. In their soluble monomeric form they hinder host immune response by recruiting, activating, and lysing human neutrophils while also promoting biofilm dissociation (*Schwartz et al., 2012*). However, self-assembly of PSMs into amyloid fibrils fortify the biofilm matrix to resist disassembly by mechanical stress and matrix degrading enzymes (*Bleem et al., 2017*). The genes encoding the core family of PSMs peptides are highly conserved and located in *psmα* operon (PSMα1–PSMα4) and *psmβ* operon (PSMβ1 and PSMβ2), and the δ-toxin is encoded within the coding sequence of RNAIII (*Peschel and Otto, 2013 Table 2*). High expression of PSMαs,~20 residues in length, increases virulence potential of methicillin-resistant *S. aureus* (*Wang et al., 2007*). Moreover, PSMα3, the most cytotoxic and lytic

PSM, enhances its toxicity to human cells upon fibrillation (*Tayeb-Fligelman et al., 2017*). Despite lower concentrations, the larger PSMβs,~44 residues in length, seem to have the most pronounced impact on biofilm structuring (*Periasamy et al., 2012*). Despite the formation of functional amyloids in *S. aureus* by PSMs, many questions remain about the intrinsic molecular mechanism by which they self-assemble and what molecular events trigger the formation of fibrillar structure from their monomeric precursor peptide. Here we apply a combination of chemical kinetic studies along with biophysical techniques to explore the relative importance of different microscopic steps involved in the mechanism of fibril formation of PSMs peptides.

## Results

### Chemical kinetics reveals different aggregation mechanisms for different PSMs

To investigate the dominating mechanism of aggregation for the individual PSMs we used chemical kinetics to analyze the aggregation of all the seven individual PSM peptides under quiescent conditions. Recently, kinetic models of protein aggregation (*Knowles et al., 2009*; *Meisl et al., 2016*) have been effectively applied to numerous model systems in biomolecular self-assembly (*Cohen et al., 2013*; *Collins et al., 2004*; *Meisl et al., 2014*). Through these models, the aggregation kinetics ascertain the rates and reaction orders of the underlying molecular events, allowing for the determination of the dominating molecular mechanism of formation of new aggregates. Aggregation kinetics of all seven PSMs peptides (PSMα1–4, PSMβ1 and 2, and δ-toxin) was monitored using Thioflavin T (ThT) fluorescence intensity (*LeVine, 1993*). For PSMα1, PSMα3, PSMβ1, and PSMβ2 reproducible aggregation curves were observed (*Figure 1a–c* and *Figure 1—figure supplement 1*), while for the rest of the PSM peptides (PSMα2, PSMα4, and δ-toxin) no reproducible aggregation was observed (*Figure 1—figure supplement 2a–c*). An increase in ThT fluorescence is observed for PSMα4 although this was not sigmoidal in shape and also not reproducible (*Figure 1—figure supplement 2b*). The timescale for the completion of aggregation differs significantly between the PSM ranging from ~1 hr for PSMα3 and up to ~70 hr for PSMα1. Furthermore, the aggregation of PSMβ1 was carried out at concentrations of microgram per milliliter compared to concentrations at milligram per milliliter for the other PSM peptides since at higher concentrations of PSMβ1 the lag-time during aggregation becomes monomer independent suggesting a saturation effect (*Figure 1—figure supplement 2d*).

To elucidate the dominating aggregation mechanism of PSMα1, PSMα3, PSMβ1, and PSMβ2, kinetic data were globally fitted at all monomeric concentrations concurrently by kinetic equations using the Amylofit interface (http://www.amylofit.ch.cam.ac.uk/fit) (*Meisl et al., 2016*). High quality global fits were achieved for all four peptides assuming a secondary nucleation mechanism for PSMα1, PSMα3, and PSMβ1, and a primary nucleation and elongation mechanism for PSMβ2. The presence of a single dominating aggregation mechanism for all four peptides is seen in the linear correlation between the half-time and the initial monomer concentration (*Meisl et al., 2016*; *Figure 1—figure supplement 3a–d*). In this simple nucleation-elongation (or linear self-assembly) model, the protein monomers form an initial nucleus with rate constant ($k_n$) and reaction order ($n_c$) which grow by elongation through the addition of monomers to the fibrils ends with rate constant ($k_+$). The secondary nucleation model additionally involves nucleus formation catalyzed by existing aggregates. In this model system, $k$ act as a combined parameter that controls the proliferation through secondary pathways with secondary process rate constant ($k_2$) and secondary pathway reaction order ($n_2$) with respect to monomer (*Meisl et al., 2016*). Secondary nucleation dominated aggregation mechanisms have previously been reported for disease-related amyloid fibrils, for example Aβ peptides (*Cohen et al., 2013*; *Meisl et al., 2014*), insulin (*Foderà et al., 2008*), α-synuclein (*Buell et al., 2014*; *Gaspar et al., 2017*), and islet amyloid poly peptide (*Ruschak and Miranker, 2007*) whereas the nucleation-elongation model has previously been linked to functional amyloids from *E. coli* and Pseudomonas (*Andreasen et al., 2019*).

In order to confirm the dominating mechanism of aggregation of the four peptides based on chemical kinetics, seeded aggregation in a regime of very low seed concentrations (nano-molar range) to monomeric concentration was conducted. This type of experiments delivers a direct means of probing the ability of fibrils to self-replicate (*Arosio et al., 2014*; *Buell et al., 2014*). The presence

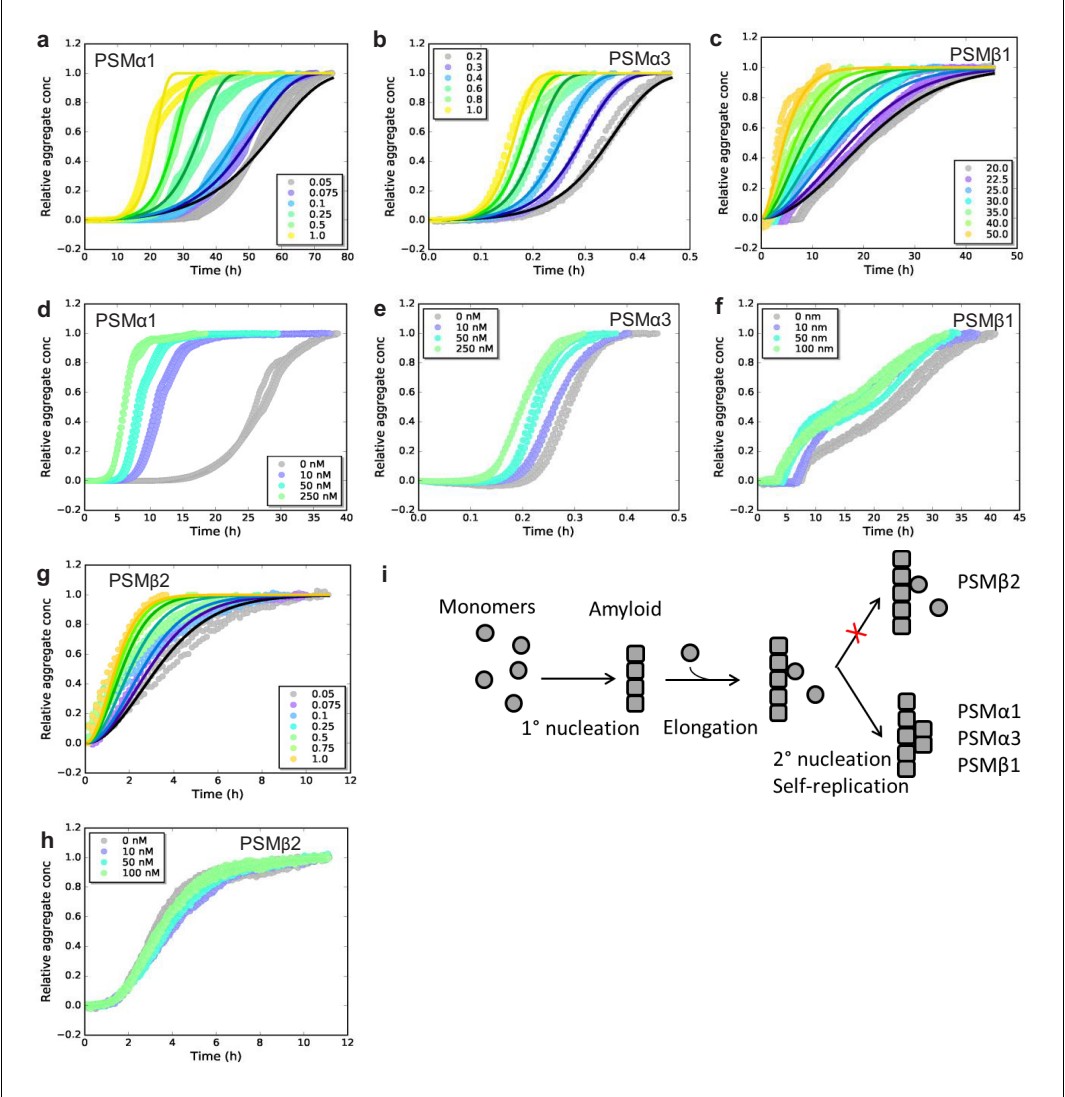

**Figure 1.** Experimental kinetic data for the aggregation of phenol-soluble modulin (PSMs) peptides from monomeric peptides. (a) Aggregation of PSMα1 (0.05–1.0 mg/mL) fitted to a secondary nucleation model. (b) Aggregation of PSMα3 (0.2–1.0 mg/mL) fitted to a secondary nucleation model. (c) Aggregation of PSMβ1 (20–50 μg/mL) fitted to a secondary nucleation model. (d) Aggregation of PSMα1 in the presence and absence of low concentrations of preformed seeds (monomers: 0.5 mg/mL, seeds: 0–250 nM). Significant effects on the rate of aggregation were observed. (e) Aggregation of PSMα3 in the presence and absence of low concentrations of preformed seeds (monomers: 0.4 mg/mL, seeds: 0–250 nM). Significant effects on the rate of aggregation were observed. (f) Aggregation of PSMβ1 in the presence and absence of low concentration of preformed seeds (monomers: 0.025 mg/mL, seeds: 0–100 nM). Significant effects on the rate of aggregation were observed. (g) Aggregation of PSMβ2 (0.05–1.0 mg/mL) fitted to a nucleation-elongation model. (h) Aggregation of PSMβ2 in the presence and absence of low concentration of preformed seeds (monomers: 0.05 mg/mL, seeds: 0–100 nM). No significant effects on the rate of aggregation are evident. (i) Schematic illustration of the microscopic steps in PSM aggregation. Monomers of PSMβ2 nucleate through primary nucleation (rate constant: $k_n$) and the aggregates grow by elongation (rate constant: $k_+$). Additionally monomers of PSMα1, PSMα3, and PSMβ1 nucleate through secondary nucleation on the surface of already existing aggregate (rate constant: $k_2$). All kinetic experiments were carried out in triplicates. Parameters from the data fitting are summarized in **Table 1**.

The online version of this article includes the following source data and figure supplement(s) for figure 1:

**Source data 1.** Kinetic data for phenol soluble peptide aggregation.
**Figure supplement 1.** Experimental kinetic raw data.
**Figure supplement 2.** Kinetic data for phenol-soluble modulin (PSM)α2, PSMα4, δ-toxin, and PSMβ1.
**Figure supplement 2—source data 1.** Kinetic data for PSMα2, PSMα4, δ-toxin and high concnetraions og PSMβ1.
**Figure supplement 3.** Half-time plots for phenol-soluble modulin (PSM)α1, PSMα3, PSMβ1, and PSMβ2.
**Figure supplement 3—source data 1.** Source data for half-time plots.
**Figure supplement 4.** Seeding with high amounts of seeds.
**Figure supplement 4—source data 1.** Seeding with high amounts of seeds for determination of elongation rates.

**Table 1.** Kinetic parameters obtained from fitting of data in *Figure 1* using the web server AmyloFit.

$n_c$ and $n_2$ are the reaction order of the primary and secondary nucleation process respectively, $k_n$ and $k_2$ are rate constants for the primary and secondary nucleation process, and $k_+$ is the rate constant for the elongation of existing fibrils.

| Parameters | PSMα1 | PSMα3 | PSMβ1 | PSMβ2 |
|---|---|---|---|---|
| Dominating mechanism | Secondary nucleation | Secondary nucleation | Secondary nucleation | Nucleation- elongation |
| Mean squared residual error (MRE) | $3.11 \times 10^{-3}$ | $1.38 \times 10^{-3}$ | $3.33 \times 10^{-3}$ | $4.75 \times 10^{-3}$ |
| $k_+k_n$ ($M^{-nc}h^{-2}$) | $6.98 \times 10^{-5}$ | 275 | $1.86 \times 10^{18}$ | 48.8 |
| $n_c$ (−) | $7.84 \times 10^{-6}$ | 0.6 | 3.92 | 0.572 |
| $k_+k_2$ ($M^{-nc}h^{-2}$) | 129 | $5.17 \times 10^6$ | $4.23 \times 10^3$ | – |
| $n_2$ (−) | $1.66 \times 10^{-3}$ | 0.123 | 0.2 | – |

of preformed fibril seeds can accelerate the aggregation process by two different mechanisms, namely elongation and surface-catalyzed secondary nucleation. The presence of low amounts of seeds eliminates the rate limiting step of primary nucleation when secondary nucleation is present but no changes in the kinetics will be observed when only primary nucleation and elongation is present as the low amounts of seed do not eliminate the need for more nuclei to be formed before the elongation process dominates. Indeed a decrease in the lag phase was observed with increasing seed concentration for PSMα1, PSMα3, and PSMβ1 (*Figure 1d–f and h*), supporting the observation that secondary nucleation is the dominating molecular mechanism for the formation of new aggregates of PSMα1, PSMα3, and PSMβ1 peptide (*Cohen et al., 2012*). The seeding effect is more clearly visible in PSMα1 in comparison to PSMα3 and PSMβ1. However, if we do the comparison through raw data we have found that almost 50% reduction in lag phase was observed in the presence of low regime of the preformed seeds in PSMβ1. Despite the very fast kinetics of PSMα3, the reduction in lag phase was significant and can be clearly observed (*Figure 1e*). The aggregation kinetics of PSMβ2 was not affected by the presence of low amounts of seeds confirming the lack of self-replication processes in the form of surface catalyzed secondary nucleation (*Figure 1h*). The general mechanism underlying formation of new aggregates from monomers of the PSM peptides from both primary and secondary pathways is shown in *Figure 1i*.

## Elongation rates differ by a factor of 1000 between fastest and slowest PSM

The relative contributions of elongation rate constant ($k_+$) were investigated in the presence of high concentration of preformed fibril seeds. The global fitting of the kinetic data yields a product of the elongation rate constant and the primary nucleation rate constant ($k_nk_+$); however, in the presence of high amounts of preformed seeds, the intrinsic nucleation process becomes negligible, and hence the aggregation under this type of experiments is only dependent on elongation of the aggregates (*Cohen et al., 2012*). The initial increase in aggregate mass was measured through linear fits to the early points of the aggregation process (*Rasmussen et al., 2019*; *Weiffert et al., 2019*; *Figure 1— figure supplement 4a, c, e and g*). The estimated elongation rate constants for PSMα1 and PSMβ2

**Table 2.** PSM peptide sequences.

The peptide sequence of the seven different PSMs from *S. aureus*.

| PSMα1 | MGIIAGIIKVIKSLIEQFTGK |
|---|---|
| PSMα2 | MGIIAGIIKFIKGLIEKFTGK |
| PSMα3 | MEFVAKLFKFFKDLLGKFLGNN |
| PSMα4 | MAIVGTIIKIIKAIIDIFAK |
| PSMβ1 | MEGLFNAIKDTVTAAINNDGAKLGTSIVSIVENGVGLLGKLFGF |
| PSMβ2 | MTGLAEAIANTVQAAQQHDSVKLGTSIVDIVANGVGLLGKLFGF |
| δ-toxin | MAQDIISTIG DLVKWIIDTVNKFTKK |

were found to be 0.2 mM/h$^2$ and 0.5 mM/h$^2$ respectively, differing by a factor of ~2, which is insignificant. Contrary to PSMα1 and PSMβ2 the estimated elongation rate constant for PSMβ1, which aggregates at very low monomeric concentrations, was found to be 0.2 µM/h$^2$ and hence a factor 1000 smaller than for PSMα1 and PSMβ2. In contrast, the elongation rate constant of PSMα3 was found to be 16.6 mM/h$^2$ exceeding the values of PSMα1 and PSMβ2 ~80- and ~35-fold, respectively. The stronger effect of elongation of PSMα3 in comparison with the other three peptides suggests that interactions with the fibrils of PSMα3 could be important during the assembly reactions in biofilm formation compared with the assembly of free monomers of other peptides into fibrils.

## Secondary structure analysis confirms α-helical structure of PSMα3 and β-sheet structure for other PSMs

The changes in secondary structure of the peptides following aggregation was monitored using benchtop circular dichroism (Jasco), synchrotron radiation circular dichroism (SRCD) spectroscopy, and Fourier transform infrared (FTIR) spectroscopy. The CD spectra (Jasco) of monomers of all PSMs peptides prior to aggregation all show double minima at 208 and 222 nm indicative of α-helical structure consistent with previous observations (*Da et al., 2017*; *Towle et al., 2016*; *Figure 2—figure supplement 1a*). Additionally, based on peak intensity at 190 and 210 nm, higher degree of α-helical secondary structure is observed for PSMα1 and PSMα3 in comparison to PSMα4 and PSMβ peptides also consistent with previous observations (*Laabei et al., 2014*). This observed increase in α-helicity in PSMα1 and PSMα3 peptides compared to PSMα4 could be due to the presence of helix stabilizing alanine residue at fifth position in PSMα1 and PSMα3 in comparison to the presence of the helix destabilizing glycine in PSMα4 (*Laabei et al., 2014*). Upon aggregation the SRCD spectrum of the peptides changes displaying a single minimum at approximately 218 nm (typical β-sheet signal) for PSMα1 and PSMα4, and at 220 nm for PSMβ1 and PSMβ2 indicative of β-sheet rich structure (*Figure 2a*). The peak positions for PSMα1 and PSMα4 are in good agreement with previous findings (*Marinelli et al., 2016*); however, we also find amyloid-like structures in the β-group of PSMs. Despite the lack of sigmoidal aggregation curves, for PSMα4 changes in the SRCD spectrum upon incubation was observed. This indicates a transition from α-helical structure to a structure with increased β-sheet content upon aggregation and is consistent with the data previously published (*Dueholm et al., 2010*; *Romero et al., 2010*). The SRCD spectrum of aggregated PSMα3 is still displaying a double minimum with minima shifted to 208 nm and 228 nm indicative of α-helical structure being present in the aggregates although this helical structure is different from that observed in the monomeric peptide. This observation is consistent with the reported cross-α-helical structure of PSMα3 aggregates (*Tayeb-Fligelman et al., 2017*). To further probe the contribution of the individual structural components to the SRCD spectra each spectrum was deconvoluted using the analysis programs Selecon3, Contin, and CDSSTR in the DichroWeb server (*Whitmore and Wallace, 2004*; *Whitmore and Wallace, 2008*; *Figure 2b* and *Figure 1—figure supplement 1*). Indeed the major structural contribution to the SRCD spectrum for PSMα3 aggregates is α-helical (~70%). For PSMα1 and PSMα4 the major structural components are β-sheet (33% and 35% respectively) and unordered structure (39% and 34% respectively). Despite the single minima indicative of predominantly β-sheet structure observed for PSMβ1 and PSMβ2 the major structural components are α-helix (35% and 40% respectively) and unordered structure (33% and 35% respectively) with less contribution from β-sheet structure (24% and 16 % respectively). Monomeric δ-toxin exhibited high degree of α-helicity structure based on CD peak intensity at 190 and 208 nm consistent with previous reports (*Laabei et al., 2014*). However, no significant structural changes were observed for PSMα2 and δ-toxin, which upon incubation at tested conditions still displayed spectra characteristic of α-helix (*Figure 2—figure supplement 1*), which is also consistent with the previous observations (*Marinelli et al., 2016*). However, we observe only one minimum at around 208 nm of δ-toxin in comparison to monomeric δ-toxin, which possess two minima at around 208 and 224 nm (*Figure 2—figure supplement 1a and b*). This is consistent with the lack of aggregation seen for these peptides by ThT fluorescence.

Consistent with the CD data, the FTIR spectra of PSMα1, PSMα4, PSMβ1, and PSMβ2 were found to be very similar to each other with a well-defined intense peak at ~1625 cm$^{-1}$ indicative of amyloid β-sheet and a minor shoulder at ~1667 cm$^{-1}$ indicative of β-turns (*Dueholm et al., 2011*; *Gasset et al., 1992*; *Zandomeneghi et al., 2004*; *Figure 2c*). The secondary structure composition of the fibrils was estimated using deconvolution of the spectra followed by conventional fitting

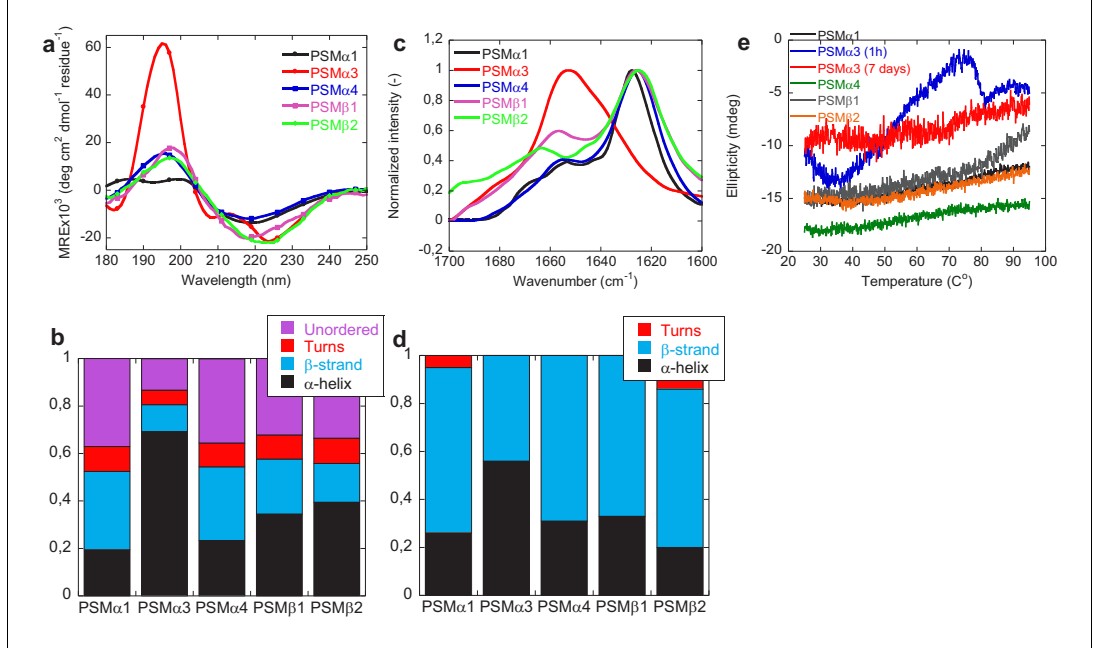

**Figure 2.** Structural comparison of fibrils formed by different phenol-soluble modulins (PSMs) variants. (**a**) Synchrotron radiation (SR) far UV-CD spectra of PSMα1, PSMα3, PSMα4, PSMβ1, and PSMβ2 fibrils recorded after 7 days of incubated samples except for PSMα3 which is recorded after 1 hr of incubated samples. (**b**) Deconvolution of the SRCD spectra of fibrils of PSM variants into the individual structural components. (**c**) Fourier transform infrared (FTIR) spectroscopy of the amide I' region (1600–1700 $cm^{-1}$) of fibrils of PSMs variants. PSMα1, PSMα4, PSMβ1, and PSMβ2 show a peak at 1625 $cm^{-1}$ corresponding to rigid amyloid fibrils. In contrast, PSMα3 shows a peak at and 1654 $cm^{-1}$ indicating α-helical structure in the fibrils. (**d**) Deconvolution of the FTIR spectra of fibrils of the PSM variants into the individual structural components. (**e**) CD (Jasco) thermal scans from 20°C to 95°C of PSMα1, PSMα3 (1 hr), PSMα3 (7 days), PSMα4, PSMβ1, and PSMβ2 fibrils.

The online version of this article includes the following source data and figure supplement(s) for figure 2:

**Source data 1.** Source data for secondary structural analysis of phenol solube modulin aggregates.

**Figure supplement 1.** CD spectra of monomeric phenol-soluble modulin (PSM) peptides and urea denaturation of PSM fibrils.

**Figure supplement 1—source data 1.** Source data for CD spectra of monomeric phenol-soluble modulin (PSM) peptides and urea denaturation of PSM fibrils.

**Figure supplement 2.** Deconvolution of Fourier transform infrared (FTIR) spectra.

**Figure supplement 2—source data 1.** Source data for the deconvolution of FTIR spectra.

program and summarized in *Figure 2d* and *Figure 2—figure supplements 2* and *Tables 3* and *4*. The percent secondary structure contribution and peaks of PSMα1 and PSMα4 from our findings are quite similar to the previous findings (*Marinelli et al., 2016*) irrespective of different experimental conditions. Additionally, a band around 1654 $cm^{-1}$ usually assigned to helical/random conformations is also noticed for all peptides, which may reflect a certain equilibrium between residual helical soluble states and a predominant aggregated assembly (*Marinelli et al., 2016*). Moreover, absence

**Table 3.** Structural contribution to FTIR spectra.

Percentage contribution of various structural components for the fibrils of PSM variants based on deconvolution of FTIR spectra along with peak position.

| Peptide | Peak position | % β-sheet | % α-helix | % β turns |
|---------|---------------|-----------|-----------|-----------|
| PSMα1 | 1626, 1653,1667 | 68.58 | 26.32 | 9.10 |
| PSMα3 | 1635, 1654 | 43.94 | 56.06 | - |
| PSMα4 | 1625, 1655 | 69.12 | 30.88 | - |
| PSMβ1 | 1625, 1656 | 66.80 | 33.2 | - |
| PSMβ2 | 1624,1644, 1664 | 66.22 | 20.05 | 13.73 |

**Table 4.** Structural contribution from deconvolution of CD spectra.

Percentage contribution of various structural components for the fibrils of PSMphenol-soluble modulin variants based on deconvolution of SRCD spectra using the DichroWeb server using the reference data setSP175 for the Selecon3, Contin, and CDSSTR analysis programs.

| Peptide | % α-helix | % β-sheet | % Turns | % Unordered |
|---------|-----------|-----------|---------|-------------|
| PSMα1 | 16.1 | 33.1 | 10.5 | 38.5 |
| PSMα3 | 69.3 | 11.3 | 6.1 | 13.2 |
| PSMα4 | 23.4 | 31.0 | 10.1 | 35.4 |
| PSMβ1 | 34.5 | 23.8 | 10.1 | 33.0 |
| PSMβ2 | 39.5 | 16.4 | 10.6 | 34.7 |

of high intensity signal ~1690 cm$^{-1}$ (characteristics of anti-parallel beta-sheet) in PSMα1 and PSMα4 suggests that β-sheets are packed in the fibrils in parallel arrangement. In good agreement with the CD data and previous reports, the FTIR spectra of PSMα3 aggregates shows significantly higher content of α-helical structure relative to other PSMs peptide fibrils as shown by a more intense band in the spectrum of at 1654 cm$^{-1}$, indicative of α-helical structure (*Kong and Yu, 2007*; *Tayeb-Fligelman et al., 2020*).

## Freshly formed PSMα3 aggregates are unstable but become stable through lateral association during maturation

The stability of the PSM aggregates was tested using CD spectroscopy (Jasco). *Figure 2e* shows the CD signal at 220 nm for fibrils (PSMα1, PSMα3, PSMα4, and PSMβ2) from 25°C to 95°C. All peptide fibrils spectra except PSMα3 indicate thermally stable β-sheet structure. Even at 95°C, there is no indication of the loss of β-sheet structure of PSMα1, PSMα4, and PSMβ2, as judged by the stable negative peak at 220 nm. However, freshly formed (1-hr old) PSMα3 fibrils are thermally unstable as a loss of structure is seen above 50°C which is consistent with the previous studies of fragments of PSMα3 (*Salinas et al., 2018*). Interestingly the lateral association of aggregates of PSMα3 upon further incubation (7 days; see *Figure 3*) renders the fibrils thermally stable and no changes in structure

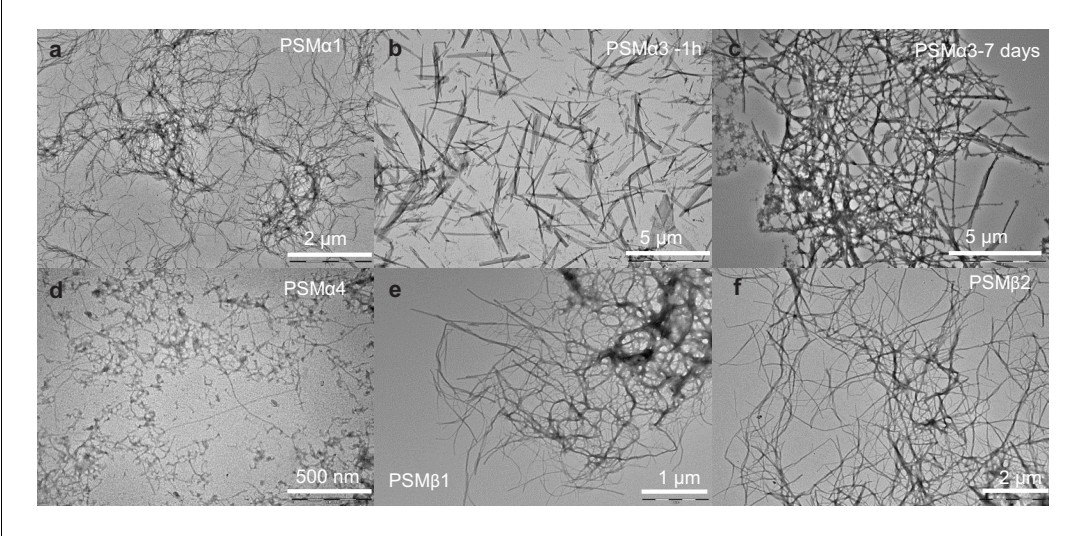

**Figure 3.** Morphology of aggregates of phenol-soluble modulin (PSMs) peptides. Transmission electron microscopic image of the end state of reaction for samples initially composed of (**a**) PSMα1 fibrils, (**b**) PSMα3 fibrils after 1 hr of incubation, (**c**) PSMα3 fibrils after 7 days of incubation, (**d**) PSMα4 fibrils, (**e**) PSMβ1 fibrils, and (**f**) PSMβ2 fibrils. Please note that scale bar changes.

The online version of this article includes the following figure supplement(s) for figure 3:

**Figure supplement 1.** Morphology of phenol-soluble modulins (PSMs) peptides.

is seen upon heating to 95°C indicating that the lateral association of the aggregates stabilizes the structure. The stability of the aggregates toward chemical denaturants was tested using urea (*Figure 2—figure supplement 1c and d*). Again, aggregates of PSMα3 (1-hr old) are the only ones susceptible toward disassembly (5–8 M urea) while no apparent effect is observed for PSMα1, PSMα4, PSMβ1, and PSMβ2 fibrils.

The morphological features of aggregates were examined using transmission electron microscopy (TEM). After 7 days of incubation, PSMα1 formed stretches of entangled fibrils (*Figure 3a*). In addition, bulky dense aggregates surrounded by a network of fibrils were observed. In contrast, PSMα3 incubated for 1 hr generated short and unbranched fibrils (*Figure 3b*), which, upon 2-day incubation, associates laterally to form stacks (*Figure 3—figure supplement 1c*) and further associates to form entangles networks of fibrils after 7 days of incubation (*Figure 3c*). Aggregates of PSMβ2 showed entangled fibrils marginally thicker and more dispersed than PSMα1 and PSMα3 (*Figure 3f*). Aggregates of PSMβ1 also formed entangled networks of fibrils (*Figure 3e*). No aggregated species could be observed for PSMα2 and δ-toxin (*Figure 3—figure supplement 1a–b*), consistent with the lack of aggregation as seen by the lack of increase in ThT fluorescence upon incubation. Interestingly, PSMα4 that lacked reproducible ThT kinetics but displayed β-sheet structure using CD and FTIR spectroscopy displays very thin fibrils visible at higher magnification with some distribution of spherical aggregates (*Figure 3d*). However, overall, these data are in good agreement with the recorded kinetics and structural data.

## PSMα1 displays promiscuous cross-seeding while other PSMs display selective cross-seeding abilities

The interplay between individual PSM peptides during formation of functional amyloids was investigated using cross-seeding experiments where the ability of aggregates of one PSM peptide to seed the aggregation of the other PSM peptides was tested. Cross-seeding experiments using 20% preformed fibril seeds of PSMα3, PSMβ1, and PSMβ2 and monomers PSMα1 were performed. It can be seen that compared to the non-seeded aggregation seeds of all the other aggregating PSM peptides PSMα3 and PSMβ1–2 accelerated the aggregation process (*Figure 4* and *Figure 4—figure supplement 1*). Similar to PSMα1, PSMβ1 aggregation is also accelerated by the presence of all the other types of preformed fibril seeds, namely PSMα1, PSMα3, and PSMβ2 seeds (*Figure 4e*). Although the resulting ThT fluorescence intensity is lower than the unseeded aggregates the lag-phase is no longer present when seeds of the other PSM peptides are present. Unlike PSMα1 and PSMβ1 the fast aggregating PSMα3 is cross-seeded by PSMα1 and PSMβ2 but not PSMβ1 (*Figure 4c*). This indicates that the cross-seeding capability of the PSM peptide aggregates is selective rather than universal among the PSM peptides. Similar to PSMα3 the aggregation of PSMβ2 is accelerated by only PSMα1 and PSMβ1 whereas PSMα3 seeded interestingly enhanced the lag phase of PSMβ2 dramatically (*Figure 4f*). Due to the presence of 20% preformed fibril seeds the initial ThT fluorescence signal is higher for the seeded experiments compared to the unseeded experiments for all the different PSM peptides. This effect is due to binding of ThT to the preformed fibril seeds at the beginning of experiment.

The PSM peptides that were not found to aggregate on their own at the conditions tested here, namely PSMα2, PSMα4, and δ-toxin, could all be induced to aggregate in the presence of different cross-seeds. Both PSMα2 and PSMα4 aggregation could be induced in the presence of PSMα1 and PSMβ1 seeds but not by PSMα3 and PSMβ2 seeds (*Figure 4b and d*). For δ-toxin aggregation could be induced by the presence of PSMα1 and PSMβ2 seeds but not PSMα3 seeds (*Figure 4g*). A slight increase in ThT fluorescence is also observed in the presence of PSMβ1 seeds but this is both very slow and a very small increase as compared to the increase seen in the presence of PSMα1 and PSMβ2 seeds.

Based on the cross-seeding analysis it is clear that the PSM peptides display selectivity in the interaction with preformed aggregates, which cannot be explained simply by the sequence similarities with in the PSM α- or β-group indicating an intricate interplay between the various PSM peptides during biofilm formation. It is also clear that PSMα1 is the most promiscuous of the PSM peptides as aggregation of PSMα1 is accelerated by all the other three PSM peptides that aggregate while also being able to accelerate the kinetics of aggregation of all the other PSM peptides, even the ones that do not aggregate on their own. Furthermore, there is no correlation between the presence or absence of secondary nucleation in the dominating aggregation mechanism for the

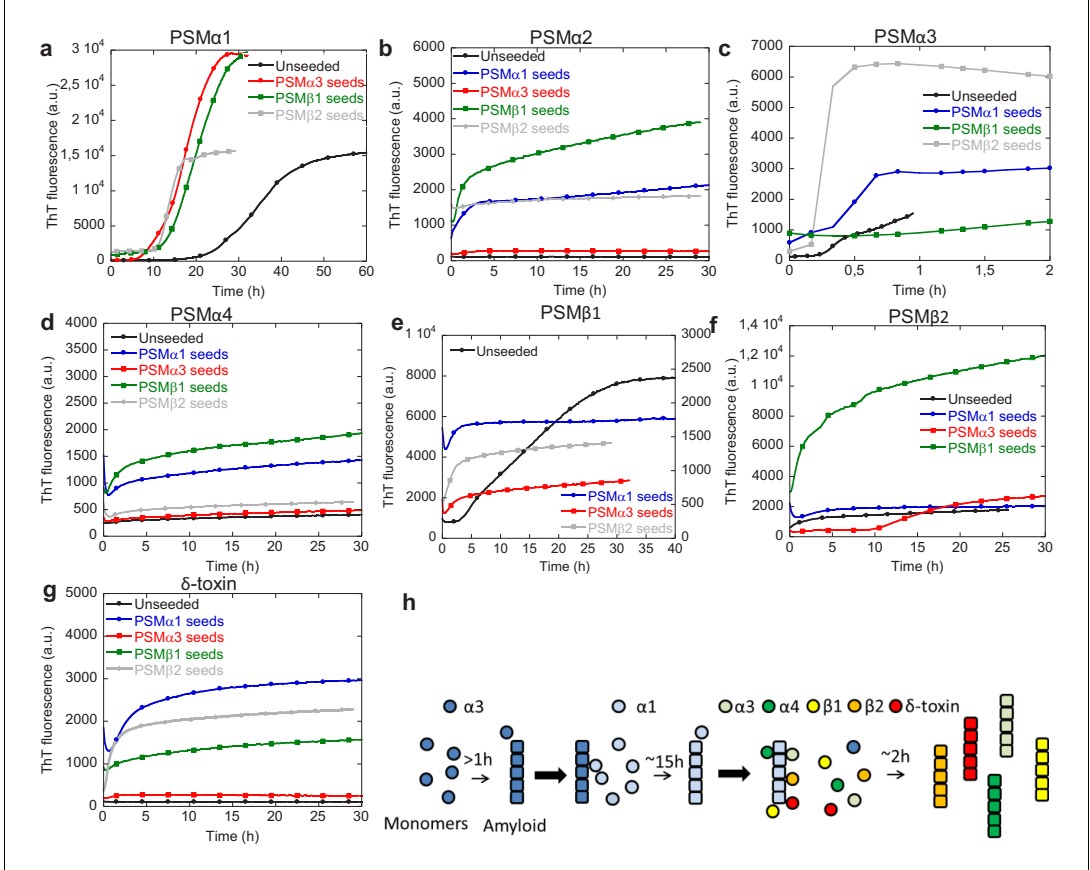

**Figure 4.** Cross-seeding phenol-soluble modulins (PSMs) variant. (**a**) Aggregation of PSMα1 (0.25 mg/mL) in the absence of seeds and in the presence of 20% (20 µM) preformed PSMα3 seeds, PSMβ1 seeds, and PSMβ2 seeds. (**b**) Aggregation of PSMα2 (0.25 mg/mL) in the absence of seeds and in the presence of 20% (20 µM) preformed PSMα1 seeds, PSMα3 seeds, PSMβ1 seeds, and PSMβ2 seeds. (**c**) Aggregation of PSMα3 (0.25 mg/mL) in the absence of seeds and in the presence of 20% (20 µM) preformed PSMα1 seeds, PSMβ1 seeds, and PSMβ2 seeds. (**d**) Aggregation of PSMα4 (0.25 mg/mL) in the absence of seeds and in the presence of 20% (20 µM) preformed PSMα1 seeds, PSMα3 seeds, PSMβ1 seeds, and PSMβ2 seeds. (**e**) Aggregation of PSMβ1 (0.025 mg/mL) in the absence of seeds and in the presence of 20% (1 µM) preformed PSMα1 seeds, PSMα3 seeds, and PSMβ2 seeds. (**f**) Aggregation of PSMβ2 (0.25 mg/mL) in the absence of seeds and in the presence of 20% (10 µM) preformed PSMα1 seeds, PSMα3 seeds, and PSMβ1 seeds. (**g**) Aggregation of δ-toxin (0.25 mg/mL) in the absence of seeds and in the presence of 20% (20 µM) preformed PSMα1 seeds, PSMα3 seeds, PSMβ1 seeds, and PSMβ2 seeds. (**h**) Schematic representation of the cross-seeding interactions between the PSM variants during biofilm formation.

The online version of this article includes the following source data and figure supplement(s) for figure 4:

**Source data 1.** Source data for the cross-seedding of phenol solube modulins.

**Figure supplement 1.** Cross-seeding of phenol-soluble modulin (PSM) peptides based on seed type.

**Figure supplement 1—source data 1.** Source data for cross-seeding of phenol soluble modulins.

**Figure supplement 2.** Aggregation propensity profiles of phenol-soluble modulin (PSM) peptides using CamSol.

**Figure supplement 2—source data 1.** Source data for aggregation propensities of phenol soluble modulins.

individual peptides and the cross-seeding capacity. PSMβ2 that aggregate through a nucleation-elongation dominated aggregation mechanism can be seeded with some (PSMα1 and PSMβ1) but not all of the peptides (PSMα3) which aggregate through a mechanism dominated by secondary nucleation. On a similar note not all the PSM peptides that aggregate through a mechanism dominated by secondary nucleation can cross-seed each other as PSMβ2 is not cross-seeded by PSMα3. We therefore suggest a model describing the delicate interplay between individual PSM peptides in the formation of biofilm where the fast aggregating PSMα3 initiates the aggregation by forming unstable fibrils. These fibrils can then accelerate the aggregation of PSMα1 that forms stable fibrils capable of accelerating the aggregation of the majority of the remaining PSM peptides (PSMα2,

PSMα3, PSMβ1, PSMβ2, and δ-toxin; *Figure 4e*). In this way the fast kinetics of PSMα3 may act as a catalyst for the whole system of aggregation of PSMs peptides during biofilm formation.

## Discussion

PSMs peptides are major determinants and play an important and diverse role in the biofilm matrix in *S. aureus* (*Peschel and Otto, 2013*). Previous studies have shown that PSMs from *S. aureus* form functional amyloids that contribute to biofilm integrity and provide resistance to disruption, which is critical to the virulence of medical device-associated infections (*Marinelli et al., 2016*; *Schwartz et al., 2012*). However, there have been no efforts to date to establish a general picture for the self-assembly of PSMs peptides that brings together all the species in the aggregation cascade. In the present study, we have conducted a combination of detailed kinetic analysis with structural and morphological analysis to gain insights into the molecular and mechanistic steps, to determine how functional amyloid of PSMs, the biofilm determinant of *S. aureus*, forms and grows. This study involves studying separately the different process involved in the aggregation reaction, i. e., initial nucleation steps, growth of fibrils, and their amplification.

Earlier computational analysis of PSMs sequences (smallest staphylococcus toxins) already suggested that the peptides of individual families might display differential self-assembly properties (*Marinelli et al., 2016*). We first determined the rates of the various microscopic steps (concentration dependent) associated with the aggregation of PSMs peptides. Our results have shown that under quiescent conditions, a dominant contribution to the formation of new aggregates is a fibril catalyzed secondary nucleation pathway that is shared by different variant of the α-PSMs family, which sustain the integrity of biofilms (*Schwartz et al., 2012*), along with the β-PSM family (PSMβ1). In contrast to this, nucleation and elongation are the only processes contributing to the aggregation of PSMβ2, since the influence of secondary process that gives rise to self-replication of aggregates is negligible for this peptide. It is remarkable that even though PSMα1 and PSMα3 possess seven identical and additional 10 similar amino acids in their sequence (*Bleem et al., 2017*), they show distinct aggregation behavior as PSMα3 aggregates approximately 50 times faster than PSMα1, which specifies that the existence of distinct residue in PSMα3 might play a significant role in lowering the energy barrier for the steps in the conversion process of monomers to fibrils as observed for Aβ peptides (*Meisl et al., 2014*). Furthermore the fibrils formed by PSMα3 was found to be initially unstable as also observed before (*Tayeb-Fligelman et al., 2017*) but upon further incubation the fibrils associate laterally to form more mature stable fibrils while fibrils of PSMα1 was stable without the need for lateral association.

In the current study at quiescent conditions no aggregation kinetics were observed for PSMα4 despite observing β-sheet structure using CD and FTIR and monitoring very thin fibrils with TEM. Previous reports on the aggregation of PSMα4 involved incubation of up to 28 days or incubation under shaking conditions (*Marinelli et al., 2016*; *Salinas et al., 2018*). As shaking conditions during aggregation is known to increase fragmentation due to shear forces (*Cohen et al., 2013*) shaking conditions were excluded in the present study. Compared to the other PSM peptides PSMα4 is the one with both the lowest solubility score and the lowest calculated aggregation propensity when computing these using the CamSol algorithm (*Sormanni et al., 2017*; *Sormanni et al., 2015*; *Figure 4—figure supplements 2* and *Table 5*). This could possibly explain the need for long incubation time and shaking conditions during aggregation.

Functional amyloids from gram-negative bacteria are mainly composed of a single protein such as CsgA in *E. coli* curli and FapC in *Pseudomonas* (*Chapman et al., 2002*; *Dueholm et al., 2010*). Along with the proteins incorporated into the functional amyloids a whole range of auxiliary proteins is expressed simultaneously. In the gram-positive bacteria *S. aureus* the functional amyloids in biofilms is made up of the different PSM peptides (*Schwartz et al., 2012*). The model suggested here accounts for the role of individual PSM peptides during formation of functional amyloids to stabilize the biofilm, hence allowing the bacteria an efficient way to form functional amyloids in a very short amount of time but at the cost of stability. The stability is later gained by the aggregation of other PSM. The most proinflammatory and cytotoxic PSMα3 (*Wang et al., 2007*) boosts the reaction of PSMα1 kinetics followed by enhancement of aggregation kinetics of rest of the PSMs peptides, which likely play a key role in stabilizing the biofilm matrix (*Schwartz et al., 2012*) and influences the biofilm development and structuring activities (*Periasamy et al., 2012*). The highly stable amyloidal

**Table 5.** Solubility score of phenol-soluble modulin peptides.
Solubility score of different peptides calculated using the Camsol web server (http://www-mvsoftware.ch.cam.ac.uk/index.php/login).

| Peptide Name | Solubility Score |
| --- | --- |
| PSMα1 | 0.826149 |
| PSMα2 | 0.883944 |
| PSMα3 | 1.282062 |
| PSMα4 | 0.022660 |
| PSMβ1 | 1.038055 |
| PSMβ2 | 0.907007 |
| δ-toxin | 1.242128 |

structures thus serve as the building blocks cementing the biofilm and creating the rigidity that can explain the resistance of amyloid-containing biofilms. Overall, we note that the rates of individual kinetic steps in the process can differ by several orders of magnitude between different variants, whereas in previous reports a vast structural diversity of amyloid-like structures have also been reported for PSM peptides (*Salinas et al., 2018*). Further, in vitro studies also confirmed that contrary to what previously thought (*Schwartz et al., 2012*), not all PSMs form amyloid structures even at higher concentrations under the conditions tested here. Moreover, cross-seeding results show that given the right conditions, all tested PSMs can aggregate into ThT-binding species.

We conclude that the outcomes presented in this article may have significant implications for understanding the aggregation process of PSMs peptides during biofilm formation. These findings indicate a molecular interplay between individual PSM peptides during accumulation of PSMs amyloid fibrils in biofilms. This also suggests an important approach for suppressing the biofilm growth of *S. aureus* as PSMs have critical role during infection and represent a promising target for anti-staphylococcal activity (*Cheung et al., 2014*). Recently potential inhibitors of Aβ aggregation in Alzheimer's and α-synuclein in Parkinson's disease have been found to inhibit self-replication by secondary nucleation being the most promising candidate (*Cohen et al., 2015*). In the case of *S. aureus* biofilm forming amyloids this could also be a potential strategy as several of the PSM peptides aggregated through a secondary nucleation dominated mechanism. This could be possible by using inhibitors of amyloid formation as numerous studies have demonstrated that inhibitors of aggregation also tend to inhibit biofilm formation (*Arita-Morioka et al., 2018*; *Arita-Morioka et al., 2015*). In the context of the development of biofilm formation, the key processes and mechanisms revealed in this study are likely to contribute to the difficulty in controlling and to understanding the role of amyloid growth as a potentially limiting factor of biofilm formation.

# Materials and methods

## Key resources table

| Reagent type (species) or resource | Designation | Source or reference | Identifiers | Additional information |
| --- | --- | --- | --- | --- |
| Peptide, recombinant protein | PSMα1 | GenScript Biotech, The Netherlands | | Formylation (N-terminal) |
| Peptide, recombinant protein | PSMα2 | GenScript Biotech, The Netherlands | | Formylation (N-terminal) |
| Peptide, recombinant protein | PSMα3 | GenScript Biotech, The Netherlands | | Formylation (N-terminal) |

*Continued on next page*

*Continued*

| Reagent type (species) or resource | Designation | Source or reference | Identifiers | Additional information |
|---|---|---|---|---|
| Peptide, recombinant protein | PSMα4 | GenScript Biotech, The Netherlands | | Formylation (N-terminal) |
| Peptide, recombinant protein | PSMβ1 | GenScript Biotech, The Netherlands | | Formylation (N-terminal) |
| Peptide, recombinant protein | PSMβ1 | GenScript Biotech, The Netherlands | | Formylation (N-terminal) |
| Peptide, recombinant protein | δ-toxin | GenScript Biotech, The Netherlands | | Formylation (N-terminal) |
| Chemical compound, drug | 2,2,2-Trifluoro-acetic acid | Sigma Aldrich | Sigma T6508 | |
| Chemical compound, drug | Thioflavin T | Sigma Aldrich | Sigma T3516 | |
| Chemical compound, drug | 1,1,1,3,3,3-Hexafluoro-2-propanol | Sigma Aldrich | Aldrich-105228 | |
| Chemical compound, drug | DMSO | Merck | CAS# 67-68-5 | |
| Software, algorithm | Amylofit | https://www.amylofit.ch.cam.ac.uk/ | | |
| Software, algorithm | Dichroweb | http://dichroweb.cryst.bbk.ac.uk/html/home.shtml | RRID:SCR_018125 | |
| Software, algorithm | CamSol | http://www-vendruscolo.ch.cam.ac.uk/.uk/camsolmethod. | | |
| Software, algorithm | OPUS 5.5 | Bruker | | |
| Other | 96-well plate, half area, polystyrene, non-binding surface | Corning | Product number 3881 | |

## Peptides and reagents

N-terminally formylated PSM peptides (>95% purity) were purchased from GenScript Biotech, The Netherlands. ThT, trifluoroacetic acid (TFA), and hexafluoroisopropanol (HFIP) were purchased from Sigma Aldrich. Dimethyl sulfoxide (DMSO) was purchased from Merck. Ultra-pure water was used for the entire study.

## Peptide pretreatment

Lyophilized PSM peptide stocks were dissolved to a concentration of 0.5 mg mL$^{-1}$ in a 1:1 mixture of HFIP and TFA followed by a 5 × 20 s sonication with 30 s intervals using a probe sonicator, and incubation at room temperature for 1 hr. The HFIP/TFA mixture was evaporated by speedvac at 1000 rpm for 3 hr at room temperature. Dried peptide stocks were stored at −80℃ prior to use.

## Preparation of samples for kinetics experiments

All kinetic experiments were performed in 96-well black Corning polystyrene half-area microtiter plates with a non-binding surface incubated at 37℃ in a Fluostar Omega plate reader (BMG Labtech,

Germany). Aliquots of purified PSMs were thawed and dissolved in DMSO to a concentration of 10 mg mL$^{-1}$ prior to use. Freshly dissolved peptides were diluted into sterile MilliQ water containing 0.04 mM ThT. To each well 100 μL of samples was added and the plate was sealed to prevent evaporation. The ThT fluorescence was measured every 10 min with an excitation filter of 450 nm and an emission filter of 482 nm at quiescent conditions. However, for PSMα3 the measurement was done every 15 s with same excitation and emission wavelength. The ThT fluorescence was followed by three repeats of each monomer concentration.

### Pre-seeded kinetic assay

Fibrils of different peptides were collected and sonicated for 3 × 10 s using a probe sonicator, at room temperature in low bind Eppendorf tubes (Axygen). Seeds were added to fresh monomer of corresponding peptide immediately before ThT measurements. In cross-seeding experiment, seeds (PSMα1 and α3, PSMβ1 and β2) were added to monomer of all other PSMs variants. ThT fluorescence was observed in the plate reader every 10 min under quiescent conditions.

### Calculation of the elongation rate constant

To estimate the rates of fibril elongation seeded aggregation with high concentration of preformed fibril seeds (20–50% of monomeric equivalents concentration) and fixed monomeric concentrations (0.25 mg/mL of PSMα1, 0.5 mg/mL of PSMα3, 0.025 mg/mL for PSMβ1, and 0.25 mg/mL for PSMβ2) was performed. The initial gradients (first 120 min for PSMα1, PSMβ1, and PSMβ2 and the first 120 s for PSMα3) of the kinetic curves were determined and plotted against the monomer concentration. Data points at higher concentration were excluded due to saturation effect of elongation.

### Far-UV circular dichroism (CD) spectroscopy

CD was performed on a JASCO-810 Spectrophotometer at 25°C, wavelength 200–250 nm with a step size of 0.1 nm, 2 nm bandwidth, and a scan speed of 50 nm/min. Samples were loaded in a 1 mm Quartz cuvette. Triplicate samples containing various peptide concentrations of each freshly dissolved peptide were pelleted and supernatant was transferred to clean sterile tube. The remaining pellet was resuspended in the same volume of MilliQ water followed by bath sonication and examined individually in far UV-CD. For each sample, the average of five scans was recorded and corrected for baseline contribution and the milliQ signal was subtracted.

### Synchotron radiation circular dichroism (SRCD) spectroscopy

The SRCD spectra of the various PSM fibrils were collected at the AU-CD beamline of the ASTRID2 synchrotron, Aarhus University, Denmark. To remove DMSO from the solution, fibrillated samples were centrifuged (13,000 rpm for 30 min), supernatants discarded, and the pellet resuspended in the same volume of MilliQ water and pellets of each sample were assayed separately. Three to five successive scans over the wavelength range from 170 to 280 nm were recorded at 25°C, using a 0.1 mm path length cuvette, at 1 nm intervals with a dwell time of 2 s. All SRCD spectra were processed and subtracted from their respective averaged baseline (solution containing all components of the sample, except the protein), smoothing with a seven pt Savitzky–Golay filter, and expressing the final SRCD spectra in mean residual ellipticity. The SRCD spectra of the individual PSM fibrils samples were deconvoluted using DichroWeb (*Whitmore and Wallace, 2004*; *Whitmore and Wallace, 2008*) to obtain the contribution from individual structural components. Each spectrum was fitted using the analysis programs Selecon3, Contin, and CDSSTR with the SP175 reference data set (*Lees et al., 2006*) and an average of the structural component contributions from the three analysis programs was used.

### Fourier transform infrared spectroscopy

Tensor 27 FTIR spectrometer (Bruker) equipped with attenuated total reflection accessory with a continuous flow of N$_2$ gas was used to collect spectra of different aliquots. Fibrillated samples were collected and centrifuged (13,000 rpm for 30 min), supernatants discarded, and the pellet resuspended in half the original volume of MilliQ water to make samples more concentrated. Of each sample 5 μL was spread on ATR crystal and let dry under nitrogen gas to purge water vapors.

Absorption spectra were recorded on the dry samples. For each spectrum 64 interferograms were accumulated with a spectral resolution of 2 $cm^{-1}$ in the range from 1000 to 3998 $cm^{-1}$ and spectra in the region 1600–1700 $cm^{-1}$ are represented. The data were processed by baseline correction and interfering signals from $H_2O$ and $CO_2$ were removed using the atmospheric compensation filter. Further, peak positions were assigned where the second order derivative had local minima and the intensity was modeled by Gaussian curve fitting using the OPUS 5.5 software. All absorbance spectra were normalized for comparative study.

### Transmission electron microscopy (TEM)

Fibrillated samples (all peptides) were collected following the ThT fibrillation kinetics assay by combining the contents of two to three wells from the plate. Five-microliter samples of all peptides were directly placed on carbon coated formvar grid (EM resolutions), allowed to adhere for 2 min, and washed with MilliQ water followed by negative staining with 2% uranyl acetate for 2 min. Further, the grids were washed twice with MilliQ water and blotted dry on filter paper. The samples were examined using a Morgagni 268 from FEI Phillips Electron microscopy, equipped with a CCD digital camera, and operated at an accelerating voltage of 80 KV.

### Fibril stability

CD (Jasco) spectra of fibrils (PSMα1, α3, and α4 and PSMβ1 and β2) were recorded from 25°C to 95°C with a step size of 0.1°C at 220 nm. Stability toward denaturants was tested by dialyzing fibrils (PSMα1, α3, and α4 and PSMβ1 and β2) containing various concentrations of urea (0–8 M) for 24 hr. One-hour incubated samples of PSMα3 fibrils were used for chemical stability.

## Acknowledgements

This work was supported by Aarhus University Research Foundation. MA is the recipient of a Starting Grant from Aarhus University Research Foundation. Furthermore, we acknowledge the award of beam time on the AU-CD beam line at ASTRID2, under project number ISA-20–1013.

## Additional information

### Funding

| Funder | Grant reference number | Author |
|---|---|---|
| Aarhus Universitets Forskningsfond | AUFF-E-2017-7-16 | Maria Andreasen |

The funders had no role in study design, data collection and interpretation, or the decision to submit the work for publication.

### Author contributions

Masihuz Zaman, Data curation, Formal analysis, Validation, Investigation, Visualization, Methodology, Writing - original draft, Writing - review and editing; Maria Andreasen, Conceptualization, Resources, Supervision, Funding acquisition, Methodology, Project administration, Writing - review and editing

### Author ORCIDs

Maria Andreasen (iD) https://orcid.org/0000-0002-6096-2995

### Decision letter and Author response

Decision letter https://doi.org/10.7554/eLife.59776.sa1
Author response https://doi.org/10.7554/eLife.59776.sa2

## Additional files

### Supplementary files
• Transparent reporting form

### Data availability

All data generated or analysed during this study are included in the manuscript and supporting files. Source data files have been provided for Figures 1, 2 and 4 in addition to Figure 1 - Figure supplement 3, 4 and 5, Figure 2 - Figure supplement 1 and 3, and Figure 4 - Figure supplement 1 and 2.

The following dataset was generated:

| Author(s) | Year | Dataset title | Dataset URL | Database and Identifier |
|---|---|---|---|---|
| Andreasen M, Zaman M | 2020 | Staphylococcus aureus phenol soluble modulin aggregation kinetics | https://doi.org/10.5061/dryad.w6m905qmx | Dryad Digital Repository, 10.5061/dryad.w6m905qmx |

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
