## [Decision Letter]

**Acceptance summary:**

Biofilm formation plays a major role in the virulence of the pathogen *Staphylococcus aureus* by mediating resistance to the host immune response. In this study, Zaman and Andreasen used biophysical techniques to analyze the basic aggregation mechanism of individual phenol-soluble modulins (PSMs), the structural scaffold of *S. aureus* biofilms. They dissected the molecular events that trigger the formation of fibrillar structure in the monomeric precursor peptides, resulting in a model for how the PSMs collectively contribute during the biofilm formation process. The insights from this study may help in the development of treatment strategies for biofilm-forming methicillin-resistant *S. aureus*.

**Decision letter after peer review:**

Thank you for submitting your article "Cross-talk between individual phenol soluble modulins in *S. aureus* biofilm enables rapid and efficient amyloid formation" for consideration by *eLife*. Your article has been reviewed by three peer reviewers, one of whom is a member of our Board of Reviewing Editors, and the evaluation has been overseen by Dominique Soldati-Favre as the Senior Editor. The following individuals involved in review of your submission have agreed to reveal their identity: Meytal Landau (Reviewer #2); Matthew Chapman (Reviewer #3).

The reviewers have discussed the reviews with one another and editors have judged that your manuscript is of interest, but additional experiments are required before it can be published. The Reviewing Editor has drafted this decision to help you prepare a revised submission. As you can see from the reviewers' comments, a better representation of the results and clear conclusion statements of what is new (identify the major take home message) should be implemented in the revised manuscript. The present data informs on the PSM mutual effects in vitro, but whether this sheds light on the actual interplay is not definitive. In addition, the technical points raised by the reviewers should be addressed.

Reviewer #1:

In this study, Zaman et al., specifically address the basic aggregation mechanism of individual phenol-soluble modulins (PSMs), the structural scaffold of the pathogen *S. aureus* biofilms. The aggregation kinetics of individual PSMs (PSMα1-4, PSMβ1-2 and δ-toxin) was determined using the Thioflavin T (ThT) assay. By fitting the aggregation kinetics into pre-defined equations, the authors deduce the aggregation mechanism of the individual PSMs. They find that PSMα1, PSMα3 and PSMβ1 aggregate via secondary nucleation (mechanism for disease related amyloid fibrils – nucleus formation catalysed by existing aggregates), while PSMβ2 aggregates via primary nucleation-elongation (mechanism linked to functional amyloids). They further analysed and compared the elongation rates in the presence of high concentrations of preformed fibril seeds (20-50%), and found that PSMα3 fibrils had the strongest effect on elongation, suggesting that interaction of PSMα3 fibrils contributes to the assembly process during biofilm formation. Secondary structural analysis by SRCD and FTIR revealed that PSMα3 aggregates have a significantly higher content of α-helical structure and are thermally unstable compared to other PSM peptide fibrils. TEM analysis of the aggregates supported the kinetics and secondary structural analysis of the individual PSMs. Finally, the authors performed cross-seeding experiments using 20% preformed fibrils to investigate the interplay between the PSMs. They show that fibrillation of the different PSMs is coordinated, enabling the rapid formation of an initially unstable amyloid structure into a very stable biofilm structure. The interplay between PSMs in biofilm formation offers new insights into the higher-order peptide-peptide interactions and may be useful for drug design towards treatment strategies of infections with biofilm-forming pathogens.

Essential revisions:

1) The idea of cross-talk between individual PSMs, the main message of the story, in the context of biofilm formation is attractive. However, the conclusion that PSMα1 seeds accelerate the aggregation of all other PSM peptides, based on data shown in Figure 4A, is overstated. The amount of ThT binding is really low and the extent of aggregation with PSMα1 seeds appears not more prominent than the aggregation of the individual PSMs (Figure 1—figure supplement 5).

2) The statement "Pre-formed seeds of PSMβ2 were able to accelerate the aggregation of PSMα1 and PSMβ1 while also inducing aggregation of PSMα2 and δ-toxin, Figure 4D", is not convincing based on the present figure display. The authors need to better display their data to convince the reader of the cooperation between individual PSMs. Except for PSMα1 being the promiscuous aggregator, it is difficult to see the interplay between the individual PSM peptides.

3) To establish a biologically relevant cross-talk, is it possible to verify the initial interaction of PSMα3 with PSMα1 by generating PSM mutants?

4) The model in Figure 4E suggests that the fast aggregating PSMα3 initiates biofilm formation by forming unstable fibrils, which accelerates the aggregation of PSMα1. Is the resulting stable fibril consisting of both PSMα3 and PSMα1?

Reviewer #2:

In the paper, the authors assessed the biophysical properties and fibrillation of seven PSM family members, which plays a major role in *S. aureus* virulence.

The main findings include:

1) PSMs show varied aggregation kinetics:

a) PSMα1, PSMα3, PSMβ1 and PSMβ2 showed reproducible ThT aggregation kinetics, while PSMα2, PSMα4 and δ-toxin did not. This partially overlaps with previous reports on PSMs fibrillation.

b) PSMα3 shows the fastest aggregation kinetics.

c) PSMs reach saturation in aggregation kinetics in different concentrations, with PSMβ1 showing saturation in relatively low concentration compared to other PSMs.

2) PSMs seeds (of the same peptide) show varied effects on aggregation lag time:

PSMα1>PSMα3>PSMβ1, while PSMβ2 shows no seeding effects. Thus,

3) PSMs show varied aggregation mechanisms:

a) A secondary nucleation mechanism for PSMα1, PSMα3 and PSMβ1, and a primary nucleation and elongation mechanism for PSMβ2.

b) PSMα1 and PSMβ2 show a similar elongation rate constant. PSMβ1, which aggregates at very low monomeric concentrations, showed a smaller elongation rate constant by a factor of 1000. In contrast, PSMα3 shows a higher elongation rate constant (few dozen fold) compared to PSMα1 and PSMβ2.

4) PSMs show varied cross-seeding effects (See notes – this is not obvious from the figures, as controls and zoom-in view into the graphs are needed).

a) PSMα1 seeds promoted self-aggregation and cross-seeded (accelerated/induced aggregation of) all other PSMs (PSMα2, PSMα3, PSMα4, PSMβ1, PSMβ2 and δ-toxin)

b) PSMα3 seeds promoted self-aggregation and cross-seeded (accelerated aggregation of) PSMα1 and PSMβ2.

c) PSMβ1 accelerated the aggregation of PSMα1 (and PSMβ2?).

d) PSMβ2 accelerated the aggregation of PSMα1 and PSMβ1, and induced aggregation of PSMα2 and δ-toxin. No effect on PSMα3 and PSMα4.

In the Conclusions the authors suggested a model in which the different properties of the PSMs allow regulation of biofilm formation and other functions in the bacteria.

In terms of impact, the PSMs play a major role in *S. aureus* virulence and are promising anti-virulence drug targets. Therefore, understanding their Interplay and properties can advance therapeutic approaches. In addition, they provide an excellent model system for understanding properties and mechanisms of functional amyloids, by providing a simplified system of naturally produced short peptides. The complexity in their fibrillation and functional properties and interplay, as also shown here, is fascinating considering their short sequences.

Essential revisions:

1) Figure 1 – In my opinion, seeding did not produce a significant effect on the rate of aggregation of PSMβ1. The effect on PSMα3 is present, but also not very significant. The differences might be related to the fibrillation lag time of the different PSMs. For PSMα1, with a long lag time, the seeding will be more significant.

2) Table 1 and figures – "MRE" needs to be explained.

3) Figure 1—figure supplement 5 – I wonder if controls at the same PSM concentration with no seeds should be added to the extrapolation.

4) Subsection “Elongation rates differ by a factor of 1000 between fastest and slowest PSM” – " The stronger effect of elongation of PSMα3 in comparison with other three peptides suggests that interactions with the fibrils increases the importance of PSMα3 fibrils in the assembly reactions compared with the assembly of free monomers of other peptides in the solution." Sentence is not completely clear.

5) Figure 2 – Solution CD is not the best way to look at secondary structures of fibrils, which are mostly insoluble. The CD spectra after incubation show that PSMβs had a higher helical content than β, while FTIR shows differently. This is probably related to the measurement in solution, which still contains high level of soluble species. At any case, this discrepancy needs to be acknowledged. Also, individual CDs for each peptide before and after Incubation will better show the trend of change (also in Figure 2—figure supplement 1B). This point becomes less critical since CD spectra of some PSMs were already shown in other papers. In that respect, you seriously need to cite, and compare results with at least some of the papers that show CD/FTIR of PSMs:

http://dx.doi.org/10.1021/acs.biochem.6b00615

https://www.sciencedirect.com/science/article/pii/S0005273614003149

https://www.nature.com/articles/srep34552

https://www.cell.com/structure/pdf/S0969-2126(19)30445-9.pdf

https://www.ncbi.nlm.nih.gov/pmc/articles/PMC5442197/

Maybe there are more…

6) The same comment goes for the ThT/TEM results. You need to cite previous publications showing that and compare.

7) Figure 2E – Specify in the legend what the y-axis represents (CD signal at 220 nm?) Is this really the right wavelength to compare? It Is only relevant If this was the major minimum in the RT/37C spectra. Since the stability of PSMα3 is specifically discussed, monitoring the spectra at 222mm is more relevant. Also, I am not sure if I understand the interpretation here. PSMα3 is stable as a helix in the monomeric and fibril forms, thus how does monitoring the secondary structure changes reflects fibril stability? If the fibrils disassemble into monomers, then it is still helical. The same goes for the Urea experiment (Figure 2—figure supplement 1D). The results in my opinion indicate on the stability of the helical structure, regardless of fibril formation. I would support the stability analyses with micrographs of the heated fibrils. (This will be a best option for the other PSMs as well).

8) The Results section indicates use synchrotron CD, but the methods also indicate regular CD (Jasco instrument). Which one is presented where?

9) In Figure 2—figure supplement 1 – methods indicate that the samples were freshly dissolved and pelleted, while the sup was transferred to a separate tube, and the pellet was resuspended. But which one was measured and shown in Figure 2—figure supplement 1 and Figure 2A? And what was the treatment to obtain the fibrils that were measured in synchrotron CD shown in Figure 2A? This should also be indicated in the legend.

10) FTIR – the Materials and methods section does not indicate clearly how was the sample treated and what was actually measured, nor the figure legend.

11) Figure 2—figure supplement 1C – what does the y-axis represent?

12) Figure 4 – controls with no seeds are needed to appreciate the effect, especially since Figure 1 doesn't necessarily shows the same concentration, and it is not in the same scale. Zoom-in view into the flat graphs is needed as well (Similar to Figure S7, but for all of them). This is a major results in this paper and needs to be made clear.

13) Concentration of the seeds should be uniformed in the units reported (concentration in nM or %) to better compared graphs in Figure 1 and Figure 4.

14) The schematic representation in Figure 4E is not intuitive (arrows need to be replaced or explained).

15) Subsection “PSMα1 display promiscuous cross-seeding while other PSMs display selective cross-seeding abilities”: " Like PSMα3, PSMβ1 is only capable of accelerating the aggregation of PSMα1". – Both also accelerated PSMβ2 if I understand correctly (hard to appreciate from the graphs, as I indicated).

16) Subsection “PSMα1 display promiscuous cross-seeding while other PSMs display selective cross-seeding abilities”: "None of the other PSM peptides were able to induce aggregation of these two PSM peptides which also do not aggregate on their own under conditions" – add except from PSMα1.

17) Discussion section: "Further, in vitro studies also confirmed that contrary to what previously thought (Schwartz et al., 2012), not all PSMs forms amyloid structures even at higher concentrations." – should be mentioned that only in the condition tested, and moreover, the cross-seeding results actually show that given the right conditions, all tested PSMs can aggregate into ThT-binding species.

Conceptual comments:

18) Is it possible that the drastic effect of PSMα1 on self- and cross-seeding lies in the preparation of the seeds themselves? Were the seeds tested by TEM? Dissolving and measuring concentration? Since this is the major result of the paper, I feel that more validations are needed for the properties of the seeds.

19) How does the primary nucleation and elongation mechanism for PSMβ2 reconciles with the cross-seeding effect by PSMα1, PSMα3 and PSMβ1?

20) In the Discussion section, PSMa3 is suggested to "..sustain the integrity of biofilms". But it is yet unclear if this peptide is actually a part of the biofilm (it was not found in a MS analysis). It is also not clear if it is present in the same vicinity as the other PSMs after secretion.

21) Since ThT/TEM/CD/FTIR measurements were already performed for most PSMs (citations and comparisons are missing here), a clear statement of what is especially novel here is required. Overall, a more focused and specific interpretation of the results is needed, while conclusions about the in-vivo setting should be tuned down.

Reviewer #3:

The authors describe in vitro aggregation studies of the phenol-soluble modulins (PSMs) from *S. aureus*. The aggregation kinetics and molecular mechanisms of PSM amyloid formation are detailed. Importantly, the manuscript also describes work on co-aggregation or cooperation between the different PSMs. The authors take the observations on how individual PSMs aggregate and speculate on how the PSMs collectively contribute during the biofilm formation process. I have some comments and suggestions that could be addressed that might strengthen the manuscript.

Essential revisions:

1) Figure 1G - PSMβ2 seems to be following the nucleation-elongation model at the concentrations tested. It would be interesting to see if this model is still followed at lower concentrations like the concentrations used of PSMβ1 (Figure 1C). Also, it would be helpful to have the units added to each of the panels in Figure 1. That might also involve altering the panels a bit so that it is clear on visual inspection which curves are seeded reactions and which are unseeded.

2) Figure 1H- Self-seeding of PSMβ2 at the low concentrations of seed almost seems to flatten the slope of the curve (or slightly increase the lag phase). According to the nucleation-elongation model, no kinetic changes should occur in presence of low amounts of seeds. Is this reproducible?

3) Figure 1—figure supplement 4 - The figure legend mentions "three" straight line plots. I think it should be four.

4) Subsection “Secondary structure analysis confirm α-helical structure of PSMα3 and β-sheet structure for other PSMs” **–** Instead of "Figure 3B" it should be "Figure 2D".

5) Figure 2—figure supplement 1D - How old were the PSMα3 fibers that were subjected to chemical denaturation? According to heat denaturation of PSMα3 fibers (Figure 2E), 7-day old fibers are more resistant to denaturation as compared fibers that are only 1 hour old.

6) Figure 4 – It would be helpful to show individual graphs of the cross-seeding reaction be shown i.e. seeded vs unseeded? This way it might be easier to compare the effect of various seeds on the aggregation kinetics.

7) Figure 4 – At 0.25mg/ml concentration the lag time for PSMβ1 becomes independent of the monomer concentration (Figure 1—figure supplement 3). However, in Figure 4, the concentration of PSMβ1 used is 0.25mg/ml, how was this done?

8) Figure 4E – The biofilm formation model suggests that PSMα3 form unstable fibrils which are the accelerated by stable PSMα1 fibrils. PSMα1 fibrils are also suggested to be accelerating the fibril formation by other PSMs. However, the data supporting this model comes from Figure 4A-D where sonicated fibers were added as seeds. Do the authors think that unsonicated fibers will also cross-seed? Can the authors teste this as in nature sonication is not possible?

[Editors' note: further revisions were suggested prior to acceptance, as described below.]

Thank you for resubmitting your work entitled "Cross-talk between individual phenol soluble modulins in *S. aureus* biofilm enables rapid and efficient amyloid formation" for further consideration by *eLife*. Your revised article has been evaluated by Dominique Soldati-Favre (Senior Editor) and a Reviewing Editor.

The manuscript has been improved but there are some remaining issues that need to be addressed before acceptance, as outlined below:

1) Regarding the significance of self-seeding – authors please insert a statement in the manuscript text like: "The seeding effect is more clearly visible in PSMα1 in comparison to other two peptides. However, if we do the comparison through raw data we have found that almost 50% reduction in lag phase was observed in presence of low regime of the preformed seeds in PSMβ1."

2) The authors should discuss in the text why many of the curves in Figure 4 do not start at 0. Is the initial ThT fluorescence due to ThT binding to the seeds or due to rapid reaction kinetics?

---

## [Author Response]

Reviewer #1:In this study, Zaman et al., specifically address the basic aggregation mechanism of individual phenol-soluble modulins (PSMs), the structural scaffold of the pathogen S. aureus biofilms. The aggregation kinetics of individual PSMs (PSMα1-4, PSMβ1-2 and δ-toxin) was determined using the Thioflavin T (ThT) assay. By fitting the aggregation kinetics into pre-defined equations, the authors deduce the aggregation mechanism of the individual PSMs. They find that PSMα1, PSMα3 and PSMβ1 aggregate via secondary nucleation (mechanism for disease related amyloid fibrils – nucleus formation catalysed by existing aggregates), while PSMβ2 aggregates via primary nucleation-elongation (mechanism linked to functional amyloids). They further analysed and compared the elongation rates in the presence of high concentrations of preformed fibril seeds (20-50%), and found that PSMα3 fibrils had the strongest effect on elongation, suggesting that interaction of PSMα3 fibrils contributes to the assembly process during biofilm formation. Secondary structural analysis by SRCD and FTIR revealed that PSMα3 aggregates have a significantly higher content of α-helical structure and are thermally unstable compared to other PSM peptide fibrils. TEM analysis of the aggregates supported the kinetics and secondary structural analysis of the individual PSMs. Finally, the authors performed cross-seeding experiments using 20% preformed fibrils to investigate the interplay between the PSMs. They show that fibrillation of the different PSMs is coordinated, enabling the rapid formation of an initially unstable amyloid structure into a very stable biofilm structure. The interplay between PSMs in biofilm formation offers new insights into the higher-order peptide-peptide interactions and may be useful for drug design towards treatment strategies of infections with biofilm-forming pathogens.Essential revisions:1) The idea of cross-talk between individual PSMs, the main message of the story, in the context of biofilm formation is attractive. However, the conclusion that PSMα1 seeds accelerate the aggregation of all other PSM peptides, based on data shown in Figure 4A, is overstated. The amount of ThT binding is really low and the extent of aggregation with PSMα1 seeds appears not more prominent than the aggregation of the individual PSMs (Figure 1—figure supplement 5).

We have altered the presentation of our results in Figure 4 to better show the acceleration of aggregation in the presence of seeds compared to the non-seeded data. The ThT signal in the plate reader assay is highly dependent on the gain setting in the plater reader. Hence a low gain setting will result in a low signal which is not equivalent to low amount of aggregation. Another factor to consider is the fact that some aggregates show low ThT signal despite the presence of high amounts of aggregates due to other factors such as lateral association which can render ThT binding sites unavailable due to steric hindrance. Hence, we are confident that despite low ThT signal for some of the aggregates our results are not overstated but do indeed reflect the molecular interactions between individual PSM peptides.

2) The statement "Pre-formed seeds of PSMβ2 were able to accelerate the aggregation of PSMα1 and PSMβ1 while also inducing aggregation of PSMα2 and δ-toxin, Figure 4D", is not convincing based on the present figure display. The authors need to better display their data to convince the reader of the cooperation between individual PSMs. Except for PSMα1 being the promiscuous aggregator, it is difficult to see the interplay between the individual PSM peptides.

We thank the reviewer for this comment as we realized that Figure 4D in its present form is not convincingly displaying our results. For further clarification, we have modified our figure display so that the figure now displays the individual PSM peptide monomers in the absence and presence of preformed fibril seeds rather than focusing on the seed type. This allows us to better display the results and hence emphasize the cooperation taking place between PSM peptides. As a consequence of this we have also modified the text describing the panels of Figure 4.

3) To establish a biologically relevant cross-talk, is it possible to verify the initial interaction of PSMα3 with PSMα1 by generating PSM mutants?

We agree with the reviewer that this would be an interesting analysis of the molecular basis of the cross-talk between the PSM peptides. However, this type of mutational study would be a complete and comprehensive study on its own and hence we consider it to be outside the scope of the current study.

4) The model in Figure 4E suggests that the fast aggregating PSMα3 initiates biofilm formation by forming unstable fibrils, which accelerates the aggregation of PSMα1. Is the resulting stable fibril consisting of both PSMα3 and PSMα1?

The fibrils formed when seeding PSMα1 with PSMα3 are formed from PMSα1 monomers since these are the only monomeric species present along with the PSMα3 seeds. However, the resulting fibrils display α-helical structural signature observed for aggregates of PSMα3. We therefore assume that the pre-formed fibrils seeds of PSMα3 template the α-helical fibrillary structure similar to that seen published in (Tayeb-Fligelman et al., 2017) formed by elongating the PSMα3 seeds with PSMα1 monomers hence we speculate that the resulting fibrils formed when cross-seeding PSMα1 with PSMα3 consists of bot peptide. However, the sable PSMα3 fibrillar aggregates observed in Figure 2E is only composed of PSMα3 since this is the only peptide present during the formation of these.

Reviewer #2:In the paper, the authors assessed the biophysical properties and fibrillation of seven PSM family members, which plays a major role in S. aureus virulence.The main findings include:1) PSMs show varied aggregation kinetics:a) PSMα1, PSMα3, PSMβ1 and PSMβ2 showed reproducible ThT aggregation kinetics, while PSMα2, PSMα4 and δ-toxin did not. This partially overlaps with previous reports on PSMs fibrillation.b) PSMα3 shows the fastest aggregation kinetics.c) PSMs reach saturation in aggregation kinetics in different concentrations, with PSMβ1 showing saturation in relatively low concentration compared to other PSMs.2) PSMs seeds (of the same peptide) show varied effects on aggregation lag time:PSMα1>PSMα3>PSMβ1, while PSMβ2 shows no seeding effects. Thus,3) PSMs show varied aggregation mechanisms:a) A secondary nucleation mechanism for PSMα1, PSMα3 and PSMβ1, and a primary nucleation and elongation mechanism for PSMβ2.b) PSMα1 and PSMβ2 show a similar elongation rate constant. PSMβ1, which aggregates at very low monomeric concentrations, showed a smaller elongation rate constant by a factor of 1000. In contrast, PSMα3 shows a higher elongation rate constant (few dozen fold) compared to PSMα1 and PSMβ2.4) PSMs show varied cross-seeding effects (See notes – this is not obvious from the figures, as controls and zoom-in view into the graphs are needed).a) PSMα1 seeds promoted self-aggregation and cross-seeded (accelerated/induced aggregation of) all other PSMs (PSMα2, PSMα3, PSMα4, PSMβ1, PSMβ2 and δ-toxin)b) PSMα3 seeds promoted self-aggregation and cross-seeded (accelerated aggregation of) PSMα1 and PSMβ2.c) PSMβ1 accelerated the aggregation of PSMα1 (and PSMβ2?).d) PSMβ2 accelerated the aggregation of PSMα1 and PSMβ1, and induced aggregation of PSMα2 and δ-toxin. No effect on PSMα3 and PSMα4.In the Conclusions the authors suggested a model in which the different properties of the PSMs allow regulation of biofilm formation and other functions in the bacteria.In terms of impact, the PSMs play a major role in S. aureus virulence and are promising anti-virulence drug targets. Therefore, understanding their Interplay and properties can advance therapeutic approaches. In addition, they provide an excellent model system for understanding properties and mechanisms of functional amyloids, by providing a simplified system of naturally produced short peptides. The complexity in their fibrillation and functional properties and interplay, as also shown here, is fascinating considering their short sequences.Essential revisions:1) Figure 1 – In my opinion, seeding did not produce a significant effect on the rate of aggregation of PSMβ1. The effect on PSMα3 is present, but also not very significant. The differences might be related to the fibrillation lag time of the different PSMs. For PSMα1, with a long lag time, the seeding will be more significant.

We agree with the reviewer that longer lag-times gives rise to a more significant reduction in the lag-time upon seeding. To determine whether or not fragmentation or secondary nucleation is active, we performed this experiment within a low regime of preformed seeds (nM range) to find generation of new aggregates. This strategy is used to distinguish fragmentation or secondary nucleation from potentially very complex and poorly understood primary nucleation processes (Cohen et al., 2012). It also confers which mechanism is dominated as preformed seeds accelerate the reaction such as it reaches completion before the equivalent reaction without preformed seeds. The seeding effect is more clearly visible in PSMα1 in comparison to other two peptides. However, if we do the comparison through raw data we have found that almost 50% reduction in lag phase was observed in presence of low regime of the preformed seeds in PSMβ1. In the absence of preformed seeds, aggregates are only generated at a slow rate during the first 8 hours (Figure 1C) of the time course of the reaction, as indicated by the constant (approximately zero) slope of the rate profile by primary or secondary nucleation. When preformed seeds are added at the beginning of the reaction, the kinetic profile reaches saturation before the corresponding reaction without preformed seeds. The rapid increase in the slope after 3 hours (Figure 1F) indicates rapid creation of new aggregates. It means that there will be more number of growing ends in presence of low amount of seeds, which accelerates the reaction of PSMβ1. The kinetic data without preformed seeds shows that primary nucleation is not rapidly creating new aggregates at this time, and by definition, the addition of seeds cannot affect primary nucleation, pinpointing the origin of the new aggregates as the effect of secondary pathways (Cohen et al., 2012).

2) Table 1 and figures – "MRE" needs to be explained.

As per reviewer’s suggestion, we have explained it in our revised manuscript.

3) Figure 1—figure supplement 5 – I wonder if controls at the same PSM concentration with no seeds should be added to the extrapolation.

We thank the reviewer for this suggestion. We have changed the layout of the data presentation in Figure 4 from focusing on seed type to now focus on the individual PSM monomeric peptide type. While changing the Figure we have also added the non-seeded aggregation data at the same concentration for all the individual PSM peptides. This allows for better comparison of the effects of the heterogeneous seeds in the cross-seeding.

4) Subsection “Elongation rates differ by a factor of 1000 between fastest and slowest PSM” – “The stronger effect of elongation of PSMα3 in comparison with other three peptides suggests that interactions with the fibrils increases the importance of PSMα3 fibrils in the assembly reactions compared with the assembly of free monomers of other peptides in the solution." Sentence is not completely clear.

We have clarified the sentence and changed it to “The stronger effect of elongation of PSMα3 in comparison with the other three peptides suggests that interactions with the fibrils of PSMα3 could be important during the assembly reactions in biofilm formation compared with the assembly of free monomers of other peptides into fibrils”.

5) Figure 2 – Solution CD is not the best way to look at secondary structures of fibrils, which are mostly insoluble. The CD spectra after incubation show that PSMβs had a higher helical content than β, while FTIR shows differently. This is probably related to the measurement in solution, which still contains high level of soluble species. At any case, this discrepancy needs to be acknowledged. Also, individual CDs for each peptide before and after Incubation will better show the trend of change (also in Figure 2—figure supplement 1B). This point becomes less critical since CD spectra of some PSMs were already shown in other papers. In that respect, you seriously need to cite, and compare results with at least some of the papers that show CD/FTIR of PSMs:http://dx.doi.org/10.1021/acs.biochem.6b00615https://www.sciencedirect.com/science/article/pii/S0005273614003149https://www.nature.com/articles/srep34552https://www.cell.com/structure/pdf/S0969-2126(19)30445-9.pdfhttps://www.ncbi.nlm.nih.gov/pmc/articles/PMC5442197/Maybe there are more…

We thank the reviewer for pointing out the missing information with regards to the sample preparation for the CD analysis. In order to eliminate a possible contribution to the CD signal of the fibrils of PSM peptides from remaining monomeric or non-aggregated species all fibril samples were centrifuged and the supernatant with remaining non-aggregated peptide were discarded before structural analysis. Hence the signal obtained from CD analysis of the PSM samples were originating from the fibrillary structures resuspended in the absence of non-aggregated species. We have clarified this in the Materials and methods section of the revised manuscript. As per reviewers request, we have also included further references in the revised manuscript in order to compare our results with those previously published.

6) The same comment goes for the ThT/TEM results. You need to cite previous publications showing that and compare.

As suggested, we have included more references in the revised manuscript in order to better compare our results to previously published results.

7) Figure 2E – Specify in the legend what the y-axis represents (CD signal at 220 nm?) Is this really the right wavelength to compare? It Is only relevant If this was the major minimum in the RT/37C spectra. Since the stability of PSMα3 is specifically discussed, monitoring the spectra at 222mm is more relevant. Also, I am not sure if I understand the interpretation here. PSMα3 is stable as a helix in the monomeric and fibril forms, thus how does monitoring the secondary structure changes reflects fibril stability? If the fibrils disassemble into monomers, then it is still helical. The same goes for the Urea experiment (Figure 2—figure supplement 1D). The results in my opinion indicate on the stability of the helical structure, regardless of fibril formation. I would support the stability analyses with micrographs of the heated fibrils. (This will be a best option for the other PSMs as well).

We thank the reviewer for pointing out the lack of information with regards to the y-axis in Figure 2E. We have added this information in the figure.

While the both the monomeric form and the aggregated form of PSMα3 are α-helical in structure the CD signal for the two species are different. In the aggregated form the minima at approx. 222 nm becomes the most predominant minima compared to the minima at 208 nm. This is probably due to a shift in the contacts stabilizing the helical structure from intramolecular one in the monomeric species to intermolecular contacts in the aggregates species. Along with this the minimum shifts to a slightly higher wavelength. We chose to monitor the changes in CD signal at 220 nm since this wavelength would report on changes to both the β-sheet structure seen for the other PSM peptides along with the changes in the α-helical spectra of PSMα3 when changing from the monomeric α-helical to the aggregated α-helical structure.

8) The Results section indicates use synchrotron CD, but the methods also indicate regular CD (Jasco instrument). Which one is presented where?

We have used both instruments i.e., synchrotron CD as well as JASCO instrument. The fibrillary structures were studied using synchrotron CD. However, JASCO CD was used to monitor spectra of monomeric peptide along with thermal and chemical stability of fibrils. We have specified in the text and figure legend which technique has been used to obtain specific results.

9) In Figure 2—figure supplement 1 – methods indicate that the samples were freshly dissolved and pelleted, while the sup was transferred to a separate tube, and the pellet was resuspended. But which one was measured and shown in Figure 2—figure supplement 1 and Figure 2A? And what was the treatment to obtain the fibrils that were measured in synchrotron CD shown in Figure 2A? This should also be indicated in the legend.

We thank the reviewer for pointing this out. We have added this in the paper and also clarified this in the figure legend.

10) FTIR – the Materials and methods section does not indicate clearly how was the sample treated and what was actually measured, nor the figure legend.

We have made the methods section more descriptive of how the samples were treated and what was measured make it easier for the readers to follow the sample preparation.

11) Figure 2—figure supplement 1C – what does the y-axis represent?

We thank the reviewer for noticing that this bit of information was missing. The y-axis represents CD ellipticity at 220 nm. We have clarified this in the revised manuscript.

12) Figure 4 – controls with no seeds are needed to appreciate the effect, especially since Figure 1 doesn't necessarily show the same concentration, and it is not in the same scale. Zoom-in view into the flat graphs is needed as well (Similar to Figure S7, but for all of them). This is a major results in this paper and needs to be made clear.

We thank the reviewer for this comment. We have added the non-seeded aggregation data to Figure 4 while also changing the layout of Figure 4 to focus on the individual PSM monomeric type.

13) Concentration of the seeds should be uniformed in the units reported (concentration in nM or %) to better compared graphs in Figure 1 and Figure 4.

We thank the reviewer for noticing this. We have added the concentration of seeds in nM/µM along with the information of the % of seeds to make it more uniform.

14) The schematic representation in Figure 4E is not intuitive (arrows need to be replaced or explained).

As per reviewer’s suggestion we have replaced the curved arrow with normal arrows to indicate which way the reaction is proceeding.

15) Subsection “PSMα1 display promiscuous cross-seeding while other PSMs display selective cross-seeding abilities”: "Like PSMα3, PSMβ1 is only capable of accelerating the aggregation of PSMα1". – Both also accelerated PSMβ2 if I understand correctly (hard to appreciate from the graphs, as I indicated).

We thank the reviewer for allowing us to specify this. As mentioned previously we have changed the way the data in Figure 4 is presented. This makes it easier to apprehend the cross-seeding capacity. As a consequence, we have also made changes to the text where the results are presented. Hence this has been clarified in the revised text.

16) Subsection “PSMα1 display promiscuous cross-seeding while other PSMs display selective cross-seeding abilities”: "None of the other PSM peptides were able to induce aggregation of these two PSM peptides which also do not aggregate on their own under conditions" – add except from PSMα1.

We thank the reviewer for catching this typo. It has been corrected in the revised manuscript.

17) Discussion section: "Further, in vitro studies also confirmed that contrary to what previously thought (Schwartz et al., 2012), not all PSMs forms amyloid structures even at higher concentrations." – should be mentioned that only in the condition tested, and moreover, the cross-seeding results actually show that given the right conditions, all tested PSMs can aggregate into ThT-binding species.

As per reviewers request we have add a comment on this in the revised manuscript.

Conceptual comments:18) Is it possible that the drastic effect of PSMα1 on self- and cross-seeding lies in the preparation of the seeds themselves? Were the seeds tested by TEM? Dissolving and measuring concentration? Since this is the major result of the paper, I feel that more validations are needed for the properties of the seeds.

We thank the reviewer for giving us the opportunity to clarify this. All seeded were prepared using the same protocol namely collecting aggregates from a plate reader experiment followed by pelleting the aggregates using centrifugation where the supernatant with any remaining non-aggregate species is removed. The aggregates are then resuspended in the original volume of buffer and sonicated using a probe sonicator. Hence any differences in the seeds are not from the preparation of the seeds but from the aggregates/seeds themselves. This protocol for seed preparation is commonly used to study seeding in protein aggregation and has not previously been observed to alter the aggregates/seeds (Ohhashi et al., 2005 and Pfamatter et al., 2017). We therefore feel confident that the effects seen during cross-seeding experiments are due to the interactions between the PSM peptides and the seeds and are not caused by differences in the seed preparation protocol.

19) How does the primary nucleation and elongation mechanism for PSMβ2 reconciles with the cross-seeding effect by PSMα1, PSMα3 and PSMβ1?

We thank the reviewer for allowing us to clarify this point. The presence or absence of secondary nucleation dictates which aggregation mechanism the dominating one during the aggregation of the peptide when starting from a homogeneous monomeric population of the peptides. Hence it only gives information on whether the surface of the aggregates formed are capable of acting as a catalyst for the production of new aggregation nuclei from monomers of the same peptide. We see no apparent connection between the dominant aggregation mechanism and the cross-seeding. Not all peptides that aggregate via a secondary nucleated dominated mechanism can cross-seed each other (PSMβ1 does not accelerate the aggregation of PSMα3). The peptides that aggregate via a secondary nucleated dominated mechanism can be cross-seeded by PSMβ2 which on its own aggregates in a primary nucleation and elongation dominated mechanism (PSMα1 aggregation is seed by PSMβ2). Furthermore, PSMβ2 cannot be cross-seeded by all the peptides that aggregate via a secondary nucleated dominated mechanism (PSMα3 does not accelerate the aggregation of PSMβ2). The same goes for the induction of aggregation in the PSMs that do not aggregate on their own (PSMβ2 does not induce aggregation in PSMα2 and PSMα4 but does induce aggregation in δ-toxin.

We have added a few lines in the revised paper to point this out.

20) In the Discussion section, PSMa3 is suggested to "..sustain the integrity of biofilms". But it is yet unclear if this peptide is actually a part of the biofilm (it was not found in a MS analysis). It is also not clear if it is present in the same vicinity as the other PSMs after secretion.

While we agree we agree with the reviewer that until now it is not clear that this peptide is actually a part of biofilm and is present in the same vicinity as the other PSMs after secretion. However, earlier studies demonstrated that *S. aureus* produces amyloid-like fibers that contribute to biofilm integrity (Schwartz et al., 2012; Marinelli et al., 2016). Further, they do not observe PSMα3 aggregates during LC-MS/MS analysis after formic acid treatment. We make a general hypothesis that those PSMs that form fibers may provide structural integrity to biofilms. However, at the same time some in vivo studies should be performed to make a solid conclusion for PSMα3 role in biofilm integrity. To address this, we have corrected it in our revised manuscript and made a more general statement by avoiding the name of specific PSMs, which play a significant role in biofilm structural integrity.

21) Since ThT/TEM/CD/FTIR measurements were already performed for most PSMs (citations and comparisons are missing here), a clear statement of what is especially novel here is required. Overall, a more focused and specific interpretation of the results is needed, while conclusions about the in-vivo setting should be tuned down.

We agree with the reviewer’s point that various studies have performed for most of the PSMs using biophysical techniques. We have incorporated more literature references in in our manuscript to relate our results to those already published. However, there have been no efforts to date to establish a general picture for the self-assembly mechanism of the individual PSMs peptides nor has the interactions between PSM during aggregation been established. We have put further emphasis on the novel results obtained here while making a more general hypothesis on how they aggregate and communicate with each other hence tuning down the conclusions on the in vivo settings.

Reviewer #3:The authors describe in vitro aggregation studies of the phenol-soluble modulins (PSMs) from S. aureus. The aggregation kinetics and molecular mechanisms of PSM amyloid formation are detailed. Importantly, the manuscript also describes work on co-aggregation or cooperation between the different PSMs. The authors take the observations on how individual PSMs aggregate and speculate on how the PSMs collectively contribute during the biofilm formation process. I have some comments and suggestions that could be addressed that might strengthen the manuscript.Essential revisions:1) Figure 1G - PSMβ2 seems to be following the nucleation-elongation model at the concentrations tested. It would be interesting to see if this model is still followed at lower concentrations like the concentrations used of PSMβ1 (Figure 1C). Also, it would be helpful to have the units added to each of the panels in Figure 1. That might also involve altering the panels a bit so that it is clear on visual inspection which curves are seeded reactions and which are unseeded.

We thank the reviewer for allowing us to elaborate on this point. We did test lower concentrations of PSMβ2 in the same range as those tested for PSMβ1. However, at concentrations below 50 µg/mL of PSMβ2 the ThT fluorescence signal become low to the point where the data becomes non-reproducible and below 30 µg/mL we see no ThT signal above the background signal. We therefore strongly believe that the aggregation models described in the paper represents the actual aggregation mechanism.

2) Figure 1H - Self-seeding of PSMβ2 at the low concentrations of seed almost seems to flatten the slope of the curve (or slightly increase the lag phase). According to the nucleation-elongation model, no kinetic changes should occur in presence of low amounts of seeds. Is this reproducible?

We agree with the reviewer that the slope of the curve for PSM β2 in the presence of law amounts of seeds is flattened slightly compare to the non-seeded aggregation curve. However, this is due to the fact that the two experiments are conducted with peptide from two different productions batches. We have consistently observed batch to batch variations in the aggregation behavior especially with the longer peptides PSMβ1 and PSMβ2. This is something which is often seen when working with synthetically produced peptides. The aggregation kinetics are reproducible within each batch and the aggregation data from different bathes can be described by the same aggregation mechanism which is why we feel confident in our results.

3) Figure 1—figure supplement 4 - The figure legend mentions "three" straight line plots. I think it should be four.

We thank the reviewer for pointing out this typo. It has been corrected as per reviewer’s suggestion in the revised manuscript.

4) Subsection “Secondary structure analysis confirm α-helical structure of PSMα3 and β-sheet structure for other PSMs” **–** Instead of "Figure 3B" it should be "Figure 2D".

As the reviewer pointed out this has been corrected in the revised manuscript.

5) Figure 2—figure supplement 1D - How old were the PSMα3 fibers that were subjected to chemical denaturation? According to heat denaturation of PSMα3 fibers (Figure 2E), 7-day old fibers are more resistant to denaturation as compared fibers that are only 1 hour old.

We used the 1 hour PSMα3 fibril samples for the chemical denaturation analysis. As per reviewer’s request we have added this information in the methods section in the revised manuscript.

6) Figure 4 – It would be helpful to show individual graphs of the cross-seeding reaction be shown i.e. seeded vs unseeded? This way it might be easier to compare the effect of various seeds on the aggregation kinetics.

We thank the reviewer for the suggestion. We have modified the representation of the data in Figure 4 from focusing on seed type to focusing on the individual PSM peptide monomers. We have also included the non-seeded aggregation data to better compare the effects of the individual seeds.

7) Figure 4 – At 0.25mg/ml concentration the lag time for PSMβ1 becomes independent of the monomer concentration (Figure 1—figure supplement 3). However, in Figure 4, the concentration of PSMβ1 used is 0.25mg/ml, how was this done?

We apologize for typological error. We have used 0.025 mg/ml (PSMβ1) for cross-seeding experiments and this has been corrected in the revised manuscript.

8) Figure 4E – The biofilm formation model suggests that PSMα3 form unstable fibrils which are the accelerated by stable PSMα1 fibrils. PSMα1 fibrils are also suggested to be accelerating the fibril formation by other PSMs. However, the data supporting this model comes from Figure 4A-D where sonicated fibers were added as seeds. Do the authors think that unsonicated fibers will also cross-seed? Can the authors teste this as in nature sonication is not possible?

We agree with the reviewer that we are looking at conditions which do not occur in vivo by adding sonication to the aggregated species. However, in order to investigate the elongation or the cross-seeding capacity of the peptides we find that this is a necessary technique to use. It has been shown that the sonication of fibrillates proteins do not alter the structure of the fibrils but only acts to break the fibrils into shorter pieces (Ohhashi et al., 2005 and Stathopulos et al., 2008). This in turn produces more growing ends from which the fibrils can elongate. We therefore strongly believe that even unsonicated fibrils are capable of acting as seeds also in a cross-seeding setting between different PSM peptides however this seeding will happen only at the growing ends and with less growing ends the seeding will happen in a slower manner than in our experiments.

[Editors' note: further revisions were suggested prior to acceptance, as described below.]

The manuscript has been improved but there are some remaining issues that need to be addressed before acceptance, as outlined below:1) Regarding the significance of self-seeding – authors please insert a statement in the manuscript text like: "The seeding effect is more clearly visible in PSMα1 in comparison to other two peptides. However, if we do the comparison through raw data we have found that almost 50% reduction in lag phase was observed in presence of low regime of the preformed seeds in PSMβ1."

We have inserted a statement similar to that suggested by the editor.

2) The authors should discuss in the text why many of the curves in Figure 4 do not start at 0. Is the initial ThT fluorescence due to ThT binding to the seeds or due to rapid reaction kinetics?

We have added the following to subsection “PSMα1 display promiscuous cross-seeding while other PSMs display selective cross-seeding abilities” to address this concern: “Due to the presence of 20% preformed fibril seeds the initial ThT fluorescence signal is higher for the seeded experiments compared to the unseeded experiments for all the different PSM peptides. This effect is due to binding of ThT to the preformed fibril seeds at the beginning of experiment.”